# DNA replication machinery prevents Rad52-dependent single-strand annealing that leads to gross chromosomal rearrangements at centromeres

Atsushi T. Onaka [1,6,8], Jie Su [1,8], Yasuhiro Katahira [1,7], Crystal Tang [1], Faria Zafar [1], Keita Aoki[2], Wataru Kagawa [3], Hironori Niki[2], Hiroshi Iwasaki [4,5] & Takuro Nakagawa [1✉]

Homologous recombination between repetitive sequences can lead to gross chromosomal rearrangements (GCRs). At fission yeast centromeres, Rad51-dependent conservative recombination predominantly occurs between inverted repeats, thereby suppressing formation of isochromosomes whose arms are mirror images. However, it is unclear how GCRs occur in the absence of Rad51 and how GCRs are prevented at centromeres. Here, we show that homology-mediated GCRs occur through Rad52-dependent single-strand annealing (SSA). The *rad52-R45K* mutation, which impairs SSA activity of Rad52 protein, dramatically reduces isochromosome formation in *rad51* deletion cells. A ring-like complex Msh2–Msh3 and a structure-specific endonuclease Mus81 function in the Rad52-dependent GCR pathway. Remarkably, mutations in replication fork components, including DNA polymerase α and Swi1/Tof1/Timeless, change the balance between Rad51-dependent recombination and Rad52-dependent SSA at centromeres, increasing Rad52-dependent SSA that forms isochromosomes. Our results uncover a role of DNA replication machinery in the recombination pathway choice that prevents Rad52-dependent GCRs at centromeres.

[1] Department of Biological Sciences, Graduate School of Science, Osaka University, 1-1 Machikaneyama, Toyonaka, Osaka 560-0043, Japan. [2] Microbial Physiology Laboratory, Department of Gene Function and Phenomics, National Institute of Genetics, 1111 Yata, Mishima, Shizuoka 411-8540, Japan. [3] Department of Chemistry, Graduate School of Science and Engineering, Meisei University, 2-1-1 Hodokubo, Hino, Tokyo 191-8506, Japan. [4] School of Life Science and Technology, Tokyo Institute of Technology, 4259 Nagatsuta, Midori-ku, Yokohama, Kanagawa 226-8503, Japan. [5] Institute of Innovative Research, Tokyo Institute of Technology, 4259 Nagatsuta, Midori-ku, Yokohama, Kanagawa 226-8503, Japan. [6]Present address: Chitose Laboratory Corporation, 2-13-3 Nogawa-honcho, Miyamae-ku, Kawasaki, Kanagawa 216-0041, Japan. [7]Present address: Department of Immunoregulation, Institute of Medical Science, Tokyo Medical University, 6-1-1 Shinjuku-ku, Tokyo 160-8402, Japan. [8]These authors contributed equally: Atsushi T. Onaka, Jie Su. ✉email: takuro4@bio.sci.osaka-u.ac.jp

It is generally believed that homologous recombination is an error-free mechanism for the repair of DNA double-strand breaks (DSBs) and collapsed replication forks, because it uses homologous intact DNA as a template. This is true for non-crossover recombination (also called gene conversion without crossover) and for recombination that occurs between allelic positions of chromosomes. However, crossover recombination and break-induced replication (BIR) can result in gross chromosomal rearrangements (GCRs) when they occur between non-allelic positions. In crossover and half-crossover recombination, the flanking regions of chromosomes are exchanged reciprocally and non-reciprocally, respectively, through endonucleolytic cleavage of joint molecules. BIR is recombination-based replication that can copy a template DNA until its end. GCRs such as translocations can cause cell death or genetic diseases including cancer[1,2]. The choice of homologous recombination pathways is crucial to maintaining genome integrity.

There are three distinct homologous recombination pathways, based on the mechanism of homologous pairing. Rad51 is the central player in the canonical pathway. Rad51 binds single-stranded DNA (ssDNA) and catalyses strand exchange with homologous duplex DNA, producing displacement-loops[3]. In both fission yeast and budding yeast, Rad52 and Rad54 are essential for Rad51-dependent recombination[4]. Rad52 loads Rad51 onto replication protein A (RPA)-coated ssDNA in yeasts, while in mammals Rad51 loading is facilitated by BRCA2, whose mutations predispose humans to breast and ovarian cancer. A Swi2/Snf2-type of chromatin remodeller, Rad54 stabilises Rad51 nucleoprotein filaments and promotes DNA strand exchange[5]. Remarkably, both yeast and mammalian Rad52 has a unique function in the second recombination pathway, which is independent of Rad51 and Rad54[6]. Rad52 promotes the annealing of complementary ssDNA molecules, single-strand annealing (SSA)[7–9]. SSA is sometimes referred to as recombination between direct repeats that results in loss of the sequence between them. However, SSA can also occur between inverted repeats when a pair of complementary ssDNAs are available. The third pathway, called microhomology-mediated end joining (MMEJ) or alternative end joining[10], uses very short homologous DNA sequences and occurs independently of Rad51 and Rad52. In addition to their roles in DNA damage repair, an increasing body of evidence suggests that SSA and MMEJ are also involved in tumorigenesis, in contrast to Rad51-dependent recombination[11–14].

Centromeres play an important role in proper chromosome segregation[15]. However, centromeres that consist of repetitive sequences are vulnerable to rearrangements[16–18]. Many organisms, including humans and fission yeast, have repetitive sequences at centromeres, while other organisms such as budding yeast have non-repetitive short centromeres. Recombination between centromere repeats can lead to the exchange of the entire short arms of acrocentric chromosomes, called Robertsonian translocation, which is the most common chromosomal abnormality observed in humans, affecting 1 in 1000 newborns[19]. Isochromosomes whose arms are mirror images of one another are found in genetic diseases such as Turner syndrome and cancer[20–22]. In *Schizosaccharomyces pombe* and *Candida albicans*, recombination between inverted repeats at centromeres results in isochromosome formation[23,24]. Loss of Rad51 (*rad51Δ*) dramatically increases both spontaneous and DSB-induced isochromosome formation[23,25]. Previously, we have shown that Rad51 and Rad54 promote non-crossover recombination between centromere repeats, thereby suppressing Mus81-dependent crossover recombination that results in isochromosome formation[26,27]. However, it is unclear how homologous pairing occurs in the GCR event independently of Rad51.

Chromatin structures protect centromeres from rearrangements[28–32]. In fission yeast, recombination occurs at centromeres in a way distinct from other chromosomal regions[32]. At centromeres, Rad51-dependent recombination predominates and other recombination pathways appear to be inhibited. As Rad51 promotes conservative non-crossover recombination[26], the choice of recombination pathways is important for suppressing GCRs. However, it is unknown how Rad51-dependent recombination predominates at centromeres.

Here, we show that Rad52-dependent SSA is the mechanism of homologous pairing that leads to centromeric GCRs. The *rad52-R45K* mutation impairs SSA activity of Rad52 protein and reduces isochromosome formation in *rad51Δ* cells. MutS homologues Msh2 and Msh3[33], and the Mus81 resolvase[34] function in the Rad52-dependent GCR pathway. To gain insights into how Rad52-dependent SSA is suppressed at centromeres, we perform a genetic screen and find that mutations in replication fork proteins, including DNA polymerase α and ε (Pol α and Pol ε), F-box protein Pof3/Dia2/Stip1, and a fork protection complex subunit Swi1/Tof1/Timeless[35–38], increase Rad52-dependent SSA at centromeres. Mutations in Pol α or Swi1 increase Rad52-dependent isochromosome formation. Collectively, these results demonstrate that DNA replication machinery plays an important role in the recombination pathway choice at centromeres, preventing Rad52-dependent SSA that results in GCRs. This study implicates Rad52-dependent SSA in GCRs, and uncovers a link between DNA replication and the recombination pathway choice at centromeres.

## Results

**Rad52 is involved in GCRs that occur in the absence of Rad51.** Rad51, Rad52, and Rad54 are essential for Rad51-dependent recombination in yeast. Rad52 also has an ability to promote SSA, independently of Rad51 and Rad54 (Fig. 1a). Rad51 and Rad54 promote non-crossover recombination between inverted repeats that are present on the opposite sides of a centromere, suppressing the formation of isochromosomes whose breakpoints are present in centromere repeats[23,26]. In *rad51* or *rad54* mutant cells, Rad52-dependent SSA might occur between the inverted repeats to produce isochromosomes. To test this possibility, we disrupted the *rad52* gene and determined the rate of spontaneous GCRs using the extra-chromosome ChL$^C$ (Fig. 1b). ChL$^C$ is derived from fission yeast chromosome 3 (chr3) and contains the entire region of the centromere 3 (cen3)[26,39]. Because ChL$^C$ is dispensable for proliferation, we can use it to detect GCRs that are otherwise lethal in haploid cells. Cells harbouring ChL$^C$ were grown in Edinburgh minimum medium supplemented with uracil and adenine (EMM + UA), and plated onto yeast nitrogen base (YNB) media: YNB + UA and YNB + UA + 5FOA, on which Leu$^+$ and Leu$^+$ Ura$^-$ colonies are formed, respectively. Leu$^+$ Ura$^-$ colonies were transferred to EMM + U plates to inspect adenine auxotrophy. We counted Leu$^+$ Ura$^-$ Ade$^-$ clones that had lost both *ura4$^+$* and *ade6$^+$* as GCR clones (Fig. 1b, GCR). Fluctuation analysis showed that *rad51Δ* strongly increased the GCR rate (Fig. 1c), as previously observed[23,26]. *rad52Δ* also increased GCR rates, probably due to the Rad52 role in Rad51-dependent recombination. However, *rad52Δ* cells exhibited lower GCR rates than *rad51Δ* cells, suggesting that Rad52 is also required for GCRs. Indeed, loss of Rad52 reduced GCR rates in *rad51Δ* cells (Fig. 1c, *rad51Δ* and *rad51Δ rad52Δ*), demonstrating that Rad52 is required for GCRs in the absence of Rad51. Rad52-dependent SSA may act as a GCR-prone backup system to retain chromosomes when Rad51 cannot repair spontaneous DNA damage. To test this, we grew the same set of the strains in yeast extract (YE) media supplemented with leucine,

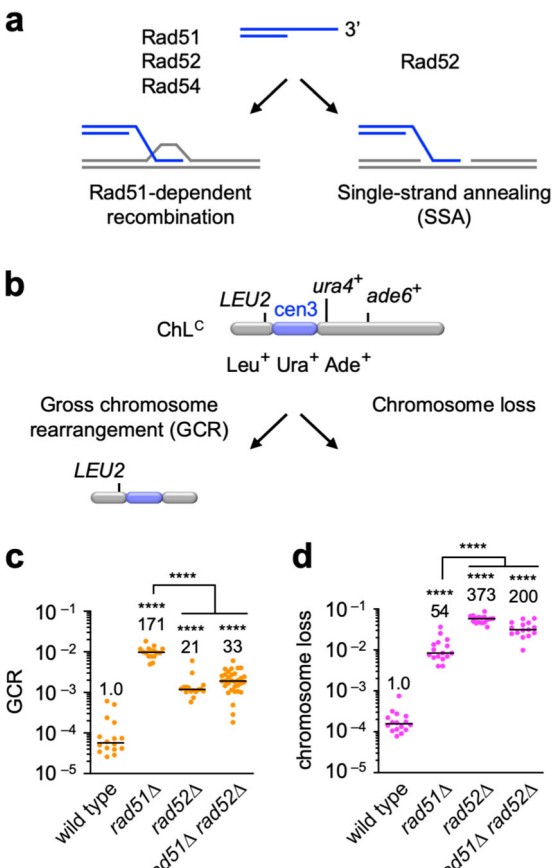

**Fig. 1 Rad52 is involved in GCRs in the absence of Rad51. a** Two pathways of homologous recombination. Rad51, Rad52, and Rad54 are involved in Rad51-dependent recombination, whereas Rad52 but not Rad51 and Rad54 is required for single-strand annealing (SSA). **b** ChL$^C$ is a derivative of Ch16, which is derived from fission yeast chr3. GCR and chromosome loss were detected by monitoring the genetic markers: *LEU2*, *ura4*$^+$, and *ade6*$^+$, which are present on ChL$^C$. **c** GCR and **d** chromosome loss rates were determined using the wild-type, *rad51Δ*, *rad52Δ*, and *rad51Δ rad52Δ* strains (TNF5369, 5411, 7493, and 7553, respectively). Each dot represents an independent experimental value obtained from an independent colony. Black lines indicate the median. Rates relative to the wild-type rate are shown on the top of each column. Statistical analyses between the wild-type and mutant strains and between the indicated pairs were performed using the two-tailed Mann–Whitney test. ****$P < 0.0001$. Source data for the graphs in **c**, **d** are available in Supplementary Data 1.

uracil, and adenine (YE3S) and counted Leu$^-$ Ade$^-$ clones resulting from ChL$^C$ chromosome loss (Fig. 1b, Chromosome loss). As previously observed[26], *rad51Δ* increased the rate of chromosome loss (Fig. 1d). Contrary to GCR, *rad52Δ* and *rad51Δ rad52Δ* cells exhibited higher rates of chromosome loss than *rad51Δ* cells, indicating that Rad52-dependent SSA plays a role in maintaining chromosomes in Rad51-deficient cells. These results demonstrate that Rad52 is involved in GCRs that occur in the absence of Rad51.

**Rad52 is specifically required for homology-mediated GCRs.** Three types of GCRs have been detected using ChL$^C$: translocation, truncation, and isochromosome formation, whose products differ in chromosome size (Fig. 2a)[26]. To determine the type(s) of GCRs that Rad52 promotes, chromosomal DNAs were prepared from parental strains and independent GCR clones, separated by pulsed-field gel electrophoresis (PFGE), and stained with

ethidium bromide (EtBr) (Fig. 2b, broad-range PFGE). In addition to chr1, chr2, and chr3, a relatively small ChL$^C$ was detected. In wild-type cells, 2 out of 34 GCR products were translocation products that were larger than the parental ChL$^C$ (Fig. 2b, sample #12; Supplementary Fig. 1a, sample #20; Table 1). We also separated the smaller GCR products by short-range PFGE (Fig. 2b), and found that all were isochromosomes (300–400 kb), but not truncation products (≤230 kb). Variations in the lengths of isochromosomes are due to different copy numbers of centromere repeats[23]. PCR analysis of GCR products recovered from agarose gels revealed that the isochromosomes retained the junctions between the central sequence cnt3 and innermost repeats imr3 (the cnt3–imr3 junctions), but not the right end of cen3 (i.e., irc3R) (Fig. 2c and Supplementary Figs. 1b and c), indicating that the breakpoints are present in centromere repeats. In *rad51Δ* cells, 2 truncates and 30 isochromosomes were detected in 32 GCR products (Fig. 2b and c; Supplementary Fig. 1; Table 1). In contrast to isochromosomes, truncation breakpoints were present either inside or outside centromeres (Table 1). Compared to *rad51Δ* cells, the proportions of truncation were increased in *rad52Δ* and *rad51Δ rad52Δ* cells (P = 0.0025 and 0.0052, respectively, two-tailed Fisher's exact test) (Table 1). The rates of each GCR type, obtained from the total GCR rate (Fig. 1c) and the proportion of each type (Table 1), showed that Rad52 is required for approximately 90% of the isochromosomes formed in *rad51Δ* cells (Fig. 2d). On the other hand, *rad52Δ* did not reduce chromosomal truncation in *rad51Δ* cells. Truncation products may be formed through telomerase activity at damage sites. These results demonstrate that Rad52 is specifically required for homology-mediated GCRs.

**SSA activity of Rad52 is required for homology-mediated GCRs.** Rad52 has DNA- and Rad51-binding domains at its N- and C-terminal regions, respectively (Fig. 3a). It has been shown in budding yeast and humans that mutating the conserved arginine residue in the DNA-binding domain (R70 in budding yeast; R55 in humans) impairs SSA but not Rad51-loading onto ssDNA[40–43]. To examine whether SSA activity of Rad52 is required for GCRs, we replaced the fission yeast arginine (R45) with lysine and determined the GCR rate. Unlike *rad52Δ* and *rad51Δ*, the *rad52-R45K* mutation did not increase GCR rates in wild-type cells (Figs. 1c and 3b), showing that *rad52-R45K* does not interfere with Rad51-dependent recombination. However, like *rad52Δ*, *rad52-R45K* reduced GCR rates in *rad51Δ* cells (Fig. 3b) and increased the proportion of truncation (P = 0.0006, two-tailed Fisher's exact test) (Table 1; Supplementary Fig. 2), indicating a specific reduction of isochromosome formation in *rad51Δ* cells (Fig. 2d). These results demonstrate that *rad52-R45K* specifically impairs the function of Rad52 to form isochromosomes. Rad52 may promote isochromosome formation even in the presence of Rad51, as *rad52-R45K* slightly but significantly reduced GCR rates in wild-type cells (Fig. 3b).

To determine whether Rad52-dependent SSA is also involved in recombination that does not result in GCRs, we detected Ade$^+$ prototrophs that are formed by gene conversion between *ade6B* and *ade6X* heteroalleles integrated at the *ura4* locus (Fig. 3c)[32]. In this arm region, both Rad51-dependent recombination and (Rad51-independent but) Rad52-dependent recombination occur[32]. While no significant effects were observed in wild-type cells, *rad52-R45K* reduced the gene conversion rate in *rad51Δ* cells (Fig. 3d), showing that Rad52-dependent SSA is required for gene conversion that occurs independently of Rad51. We also examined whether *rad52-R45K* affects sensitivity to the topoisomerase inhibitor camptothecin (CPT), which induces DSBs during DNA replication (Fig. 3e). While no obvious effects were observed

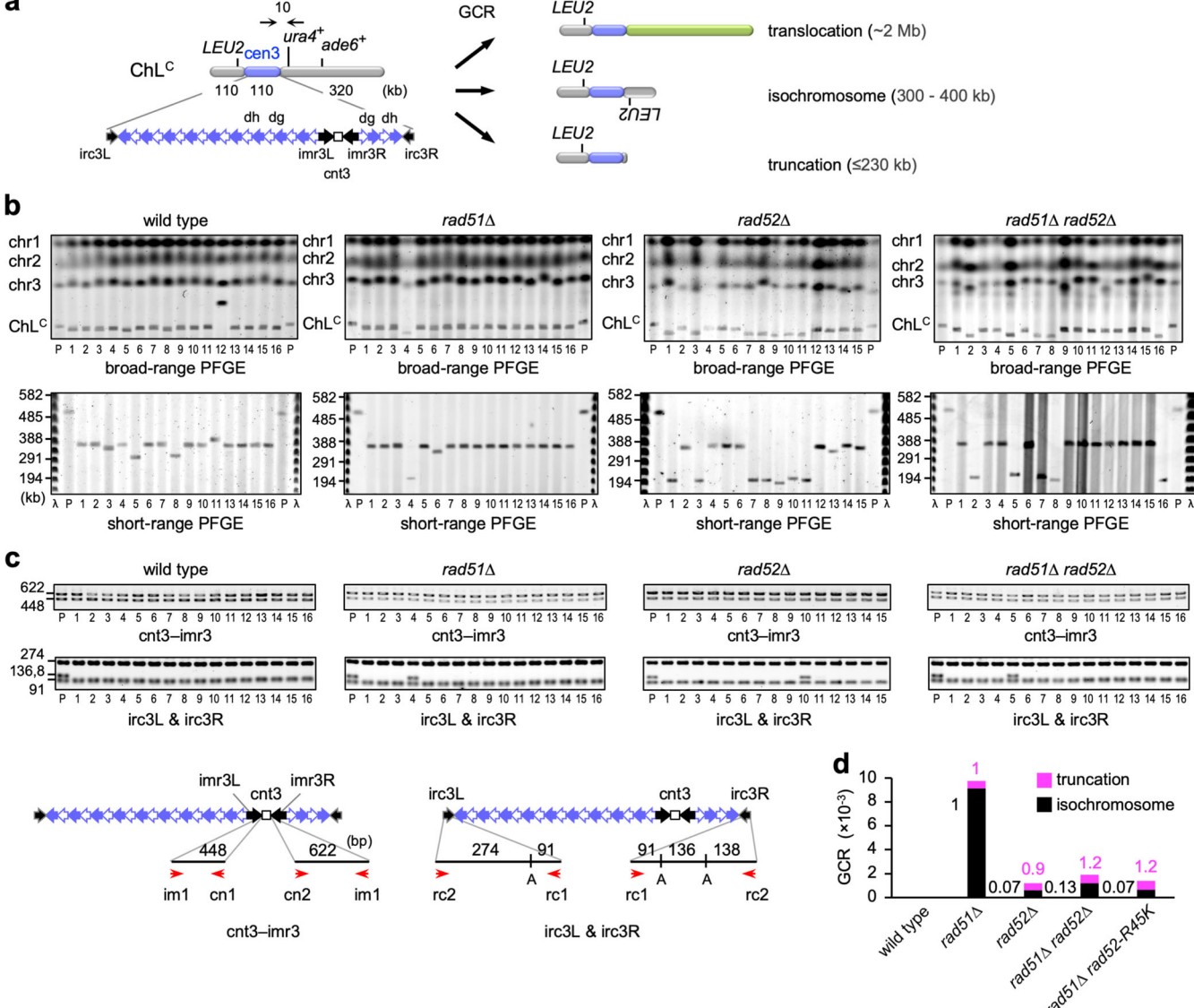

**Fig. 2 Rad52 is required for isochromosome formation but not truncation. a** Three types of GCRs (translocation, isochromosome formation, and truncation) can be detected using ChL^C. The three can be differentiated by length. **b** Chromosomal DNAs from parental strains (P) and independent GCR clones of wild type, *rad51Δ*, *rad52Δ*, and *rad52Δ rad51Δ* were separated by broad- and short-range PFGE and stained with EtBr (top and bottom rows, respectively). Positions of chr1, chr2, chr3, and ChL^C (5.7, 4.6, ~3.5, and 0.5 Mb, respectively) in the parental strains are indicated on the left side of the broad-range PFGE panels (top row). Numbers on the left of the short-range PFGE panels (bottom row) indicate sizes of λ ladders (Promega). **c** PCR analysis of GCR products recovered from agarose gels. Both sides of cnt3–imr3 junctions were amplified and resolved by standard agarose gel electrophoresis (cnt3–imr3). Amplified irc3R and irc3L regions were digested with ApoI (irc3L & irc3R) prior to the electrophoresis. Positions of primers (red arrows) and ApoI sites are indicated at the bottom. A, ApoI. **d** Rates of truncation (magenta) and isochromosome formation (black). Rates relative to that of the *rad51Δ* strain are indicated. Uncropped images of the gels presented in **b**, **c** are shown in Supplementary Fig. 7. Source data for the graph in **d** are available in Supplementary Data 1.

**Table 1 Three types of GCRs detected in this study.**

|  | Translocation | Truncation* | Isochromosome | Total GCRs |
|---|---|---|---|---|
| wild type | 2 (6%) | 0 [0] | 32 (94%) | 34 |
| *rad51Δ* | 0 | 2 [1] (6%) | 30 (94%) | 32 |
| *rad52Δ* | 0 | 7 [6] (47%) | 8 (53%) | 15 |
| *rad51Δ rad52Δ* | 0 | 12 [9] (37%) | 20 (63%) | 32 |
| *rad52-R45K* | 0 | 0 [0] | 15 (100%) | 15 |
| *rad51Δ rad52-R45K* | 0 | 8 [5] (53%) | 7 (47%) | 15 |

Percentages of each type of GCRs are shown in ().
*The number of truncation products whose breakpoints are present in centromere repeats are show in [].

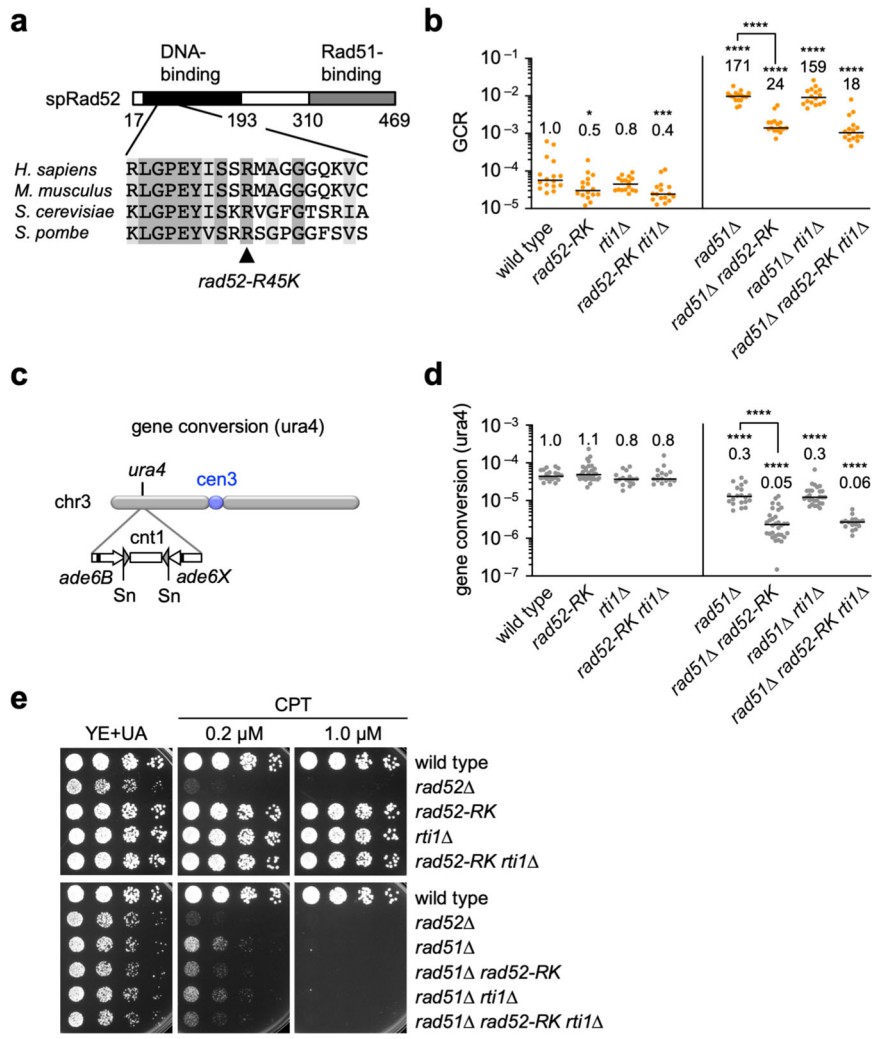

**Fig. 3 The rad52-R45K mutation reduces GCRs. a** The rad52-R45K mutation site is located in the DNA-binding domain of Rad52. spRad52, S. pombe Rad52. **b** GCR rates of the wild-type, rad52-R45K, rti1Δ, rad52-R45K rti1Δ, rad51Δ, rad51Δ rad52-R45K, rad51Δ rti1Δ, and rad51Δ rad52-R45K rti1Δ strains (TNF5369, 6599, 6707, 7879, 5411, 6616, 6725, and 7886, respectively). **c** The ade6B and ade6X repeats integrated at the ura4 locus of chr3 are illustrated[26]. Sn, SnaBI. **d** Gene conversion rates at the ura4 locus in the wild-type, rad52-R45K, rti1Δ, rad52-R45K rti1Δ, rad51Δ, rad51Δ rad52-R45K, rad51Δ rti1Δ, and rad51Δ rad52-R45K rti1Δ strains (TNF3631, 5995, 5389, 7878, 3635, 6021, 5427, and 7890, respectively). *P < 0.05; ***P < 0.001; ****P < 0.0001; ns, non-significant P > 0.05. **e** Exponentially growing cells of the strains in **d** and the rad52Δ (TNF3643) strain were 5-fold serially diluted in water and spotted onto YE + UA plates supplemented with CPT. Source data for the graphs in **b**, **d** are available in Supplementary Data 1.

in wild-type cells, rad52-R45K increased CPT sensitivity in rad51Δ cells. These data demonstrate that Rad52-dependent SSA acts as a backup system to repair endogenous and exogenous DNA damage in the absence of Rad51.

In budding yeast, the Rad52 paralog, Rad59, also shows SSA activity[42,44]. Thus, we postulated that the Rad52 paralog in fission yeast, Rti1, is also involved in GCRs. However, loss of Rti1 did not significantly affect GCR rates in wild-type, rad52-R45K, rad51Δ, or rad51Δ rad52-R45K cells (Fig. 3b). In addition, rti1Δ caused no significant effects on gene conversion and CPT sensitivity (Fig. 3d and e). These results show that Rti1 is not essential for GCRs, gene conversion, or DNA damage repair.

To determine the effect of rad52-R45K on the biochemical activity of Rad52, we expressed C-terminally Flag-tagged Rad52 in E. coli and purified the recombinant protein using anti-Flag antibodies and an anion exchange column (Fig. 4a) (see the "Methods" section). The Flag-tag did not interfere with Rad52 function, as the yeast strain expressing the Flag-tagged rad52 gene in place of the wild-type gene was no more sensitive to CPT than the wild-type strain (Supplementary Fig. 3a). First, we performed

gel mobility shift assays to evaluate the ssDNA-binding activity of Rad52 (Fig. 4b). Rad52 was incubated with $^{32}$P-labelled ssDNAs, and the complexes were resolved by non-denaturing polyacrylamide gel electrophoresis (PAGE). ssDNAs stacked in the well increased as a function of Rad52 concentrations (Fig. 4b and c), indicating the formation of Rad52-ssDNA complexes. rad52-R45K partially impaired ssDNA-binding activity, indicated by the reduced amount of ssDNAs in the well compared to wild-type Rad52. Next, we performed in vitro SSA assays (Fig. 4d) as previously described[45,46]. Radiolabelled ssDNAs were added to a mixture of Rad52 and unlabelled complementary ssDNAs to initiate the annealing reaction. After the indicated periods of time, DNAs were purified and resolved by non-denaturing PAGE (Fig. 4e). In the presence of Rad52, double-stranded DNAs (dsDNAs) increased as a function of incubation time, and nearly 80% of the labelled ssDNAs became dsDNAs within 5 min (Fig. 4f). Essentially, no dsDNAs were detected in the mock reaction, demonstrating that Rad52 is essential for SSA. rad52-R45K severely impaired SSA. At 1 min, the proportions of dsDNAs were 42.4% ± 5.0 and 4.9% ± 1.7 (mean ± s.d.) for Rad52

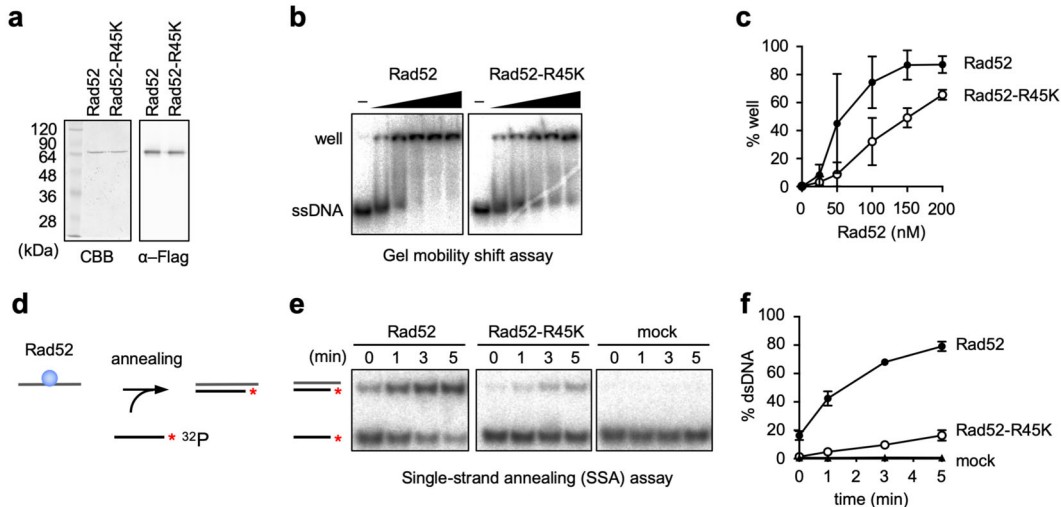

**Fig. 4 The *rad52-R45K* mutation impairs the SSA activity of the Rad52 protein. a** Purified Rad52 and Rad52-R45K proteins were separated by 12% SDS-PAGE and stained with CBB or immunostained using anti-Flag antibody (α–Flag). **b** Gel mobility shift assays. $^{32}$P-labelled Oligo211 (48 nt, 2 nM in DNA molecules) and Rad52 (0, 25, 50, 100, 150, and 200 nM) were incubated at 30 °C for 10 min. After addition of loading buffer, the mixture was resolved by 10% non-denaturing PAGE. **c** Percentages of well signals in whole-lane signals. The mean and s.d. of three independent experiments are shown. **d** Scheme of Rad52-mediated SSA assays. The asterisk represents the $^{32}$P-label. **e** SSA assays. After incubating Rad52 (1.35 nM) with Oligo508 (53 nt, 0.4 nM in DNA molecules) for 10 min at 30 °C, $^{32}$P-labelled Oligo211 (48 nt, 0.3 nM in DNA molecules) was added. After the indicated periods of time, DNAs were purified and resolved by 10% non-denaturing PAGE. **f** Percentages of dsDNA signals in whole-lane signals. Uncropped images of the gels presented in **b**, **e** are shown in Supplementary Fig. 8. Source data for the graphs in **c**, **f** are available in Supplementary Data 1.

and Rad52-R45K, respectively. Similar results were observed when ssDNAs were coated with RPA, although the overall annealing efficiency was reduced (Supplementary Figs. 3b and c). Collectively, these results demonstrate that Rad52-dependent SSA activity is required for homology-mediated GCRs.

**Msh2–Msh3 and Mus81 act in the Rad52-dependent GCR pathway.** MutS and MutL homologues are involved in DNA mismatch repair (MMR) in eukaryotes (Fig. 5a). Msh2–Msh3 and Msh2–Msh6 heterodimers form ring-like complexes that detect DNA loops and mismatches, respectively, and recruit Mlh1 to repair those replication errors. Msh2–Msh3 also binds joint molecules and is involved in SSA[33,47]. To test their involvement in Rad52-dependent GCRs, we disrupted these MMR genes (Fig. 5a). In wild-type cells, neither *msh2Δ* nor *msh3Δ* significantly affected GCR rates. However, in *rad51Δ* cells, *msh2Δ* and *msh3Δ* reduced GCR rates. On the other hand, neither *msh6Δ* nor *mlh1Δ* significantly affected GCR rates in both wild-type and *rad51Δ* cells ($P > 0.05$, two-tailed Mann–Whitney test). These results show that Msh2–Msh3 is specifically involved in GCRs. *msh2Δ* and *rad52-R45K* did not additively reduce GCR rates in *rad51Δ* cells, suggesting that Msh2 and Rad52 act in the same GCR pathway. It is likely that Msh2–Msh3 plays a supplementary role in Rad52-dependent GCRs, because, in wild-type cells, neither *msh2Δ* nor *msh3Δ* significantly reduced GCR rates ($P = 0.27$ and $0.47$, respectively) in contrast to *rad52-R45K* ($P = 0.014$, two-tailed Mann–Whitney test). In *rad51Δ* cells, *msh2Δ* and *msh3Δ* reduced GCR rates less effectively than *rad52-R45K* ($P = 0.0014$ and $0.0025$, respectively). We also noticed that *msh2Δ* increased GCR rates in *rad51Δ rad52-R45K* cells ($P < 0.0001$). It is possible that replication errors accumulated by *msh2Δ* causes GCRs in *rad51Δ rad52-R45K* cells.

The Mus81 endonuclease[27,48–50] might cleave joint molecules formed by SSA as half-crossovers (Fig. 5b). As previously observed[26], *mus81Δ* reduced GCR rates in *rad51Δ* cells, indicating the involvement of Mus81 in GCRs. Notably, *mus81Δ* and *rad52-R45K* did not additively reduce GCR rates

in *rad51Δ* cells, suggesting that joint molecules produced by Rad52-dependent SSA are cleaved by the Mus81 endonuclease to form isochromosomes.

We also examined whether Msh2 and Mus81 are also required for gene conversion at the *ura4* locus (Fig. 5c). Intriguingly, unlike *rad52-R45K*, neither *msh2Δ* nor *mus81Δ* reduced gene conversion in *rad51Δ* cells, showing that Msh2 and Mus81 are dispensable for gene conversion. Together, these results suggest that the Msh2–Msh3 ring-like complex and the Mus81 endonuclease specifically function in the Rad52-dependent GCR pathway.

**Replication machinery prevents Rad52-dependent SSA at centromeres.** At fission yeast centromeres, Rad51-dependent recombination predominates and Rad52-dependent SSA hardly occurs[32]. To prevent GCRs, Rad52-dependent SSA may be actively suppressed at centromeres. To identify factors that suppress Rad52-dependent SSA, we treated *rad54Δ* cells with nitrous acid to introduce random mutations[51] and screened for the clones that exhibited elevated levels of gene conversion at cen1 (Fig. 6a). Out of the 7400 clones examined, three reproducibly showed increased levels of Ade+ recombinants (Fig. 6b; Supplementary Fig. 4a). Two of the three clones exhibited temperature-sensitive growth defects. After several rounds of backcrossing with the wild-type strain, we introduced a genomic DNA library into the two temperature-sensitive clones, and found that *spb70* and *pof3* genes complemented the growth defects. Sanger sequencing of each genomic locus revealed that they had *spb70-G529D* and *pof3-L148R* mutations, respectively. We performed whole-genome sequencing of the remaining clone, and found the *pol1-R961K* mutation that alters an evolutionarily conserved residue in the catalytic domain of DNA Pol α (alpha) (Fig. 6c). Expression of the corresponding wild-type gene reduced Ade+ recombinants (Supplementary Fig. 4a), indicating that *pol1-R961K*, *spb70-G529D*, and *pof3-L148R* are the mutations that increased centromere recombination in *rad54Δ* cells. Pol1 and Spb70/Pol12 are catalytic and non-catalytic subunits of Pol α, respectively[35]. Pof3/Dia2/

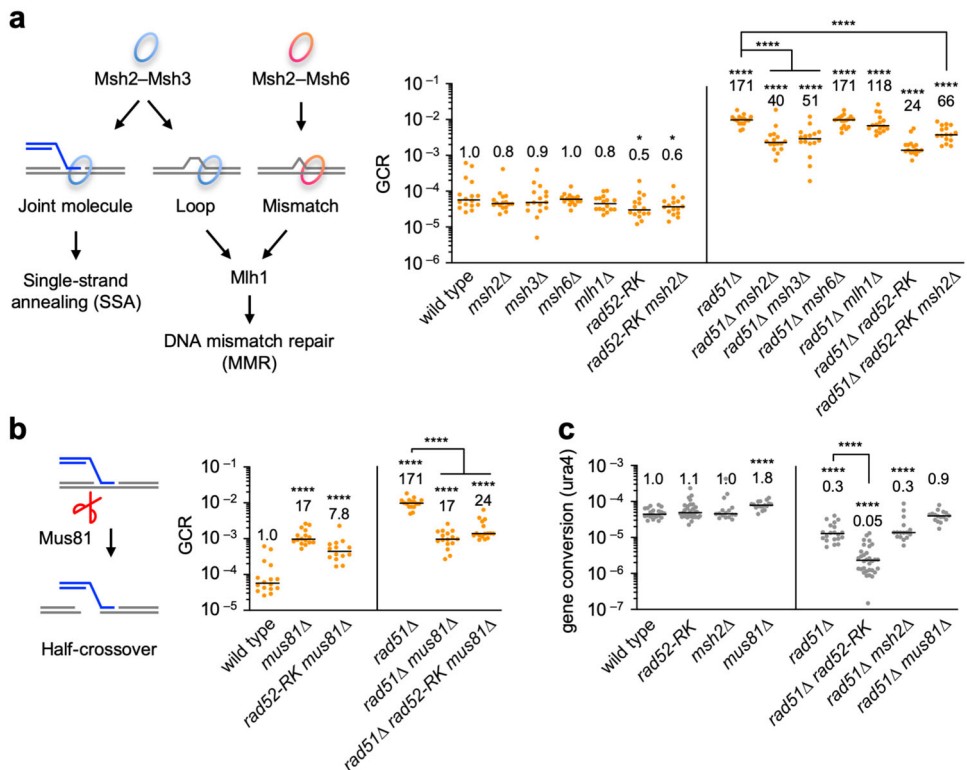

**Fig. 5 Msh2, Msh3, and Mus81 are involved in the Rad52-dependent GCR pathway. a** GCR rates of the wild-type, msh2Δ, msh3Δ, msh6Δ, mlh1Δ, rad52-R45K, rad52-R45K msh2Δ, rad51Δ, rad51Δ msh2Δ, rad51Δ msh3Δ, rad51Δ msh6Δ, rad51Δ mlh1Δ, rad51Δ rad52-R45K, and rad51Δ rad52-R45K msh2Δ strains (TNF5369, 6618, 6867, 6869, 6620, 6627, 6599, 5411, 6649, 7081, 6908, 6651, 6616, and 6697, respectively). The roles of Msh2–Msh3, Msh2–Msh6, and Mlh1 in single-strand annealing (SSA) and DNA mismatch repair (MMR) are illustrated on the left side of the graph. **b** GCR rates of the wild-type, mus81Δ, rad52-R45K mus81Δ, rad51Δ, rad51Δ mus81Δ, and rad51Δ rad52-R45K mus81Δ strains (TNF5369, 5669, 6614, 5411, 5974, and 6648, respectively). The role of Mus81 endonuclease in half-crossover formation is illustrated on the left of the graph. **c** Gene conversion rates at the ura4 locus in the wild-type, rad52-R45K, msh2Δ, mus81Δ, rad51Δ, rad51Δ rad52-R45K, rad51Δ msh2Δ, and rad51Δ mus81Δ strains (TNF3631, 5995, 6128, 6518, 3635, 6021, 6136, and 6569, respectively). *P < 0.05; ****P < 0.0001. Source data for the graphs in **a–c** are available in Supplementary Data 1.

Stip1 is the F-box protein associated with the replication machinery and is required to unload the replicative Cdc45-Mcm2-7-GINS (CMG) helicase from the DNA[37,38,52].

To determine whether Pol1 suppresses Rad52-dependent SSA at centromeres, we performed detailed analyses of recombination at cen1. Loss of Rad51, Rad54, or Rad52 equally and severely reduced recombination (Fig. 6d), demonstrating that Rad51-dependent recombination predominates and Rad52-dependent SSA hardly occurs at centromeres. The pol1-R961K mutation increased recombination in rad51Δ as well as rad54Δ cells but not in wild-type or rad52Δ cells, showing that Pol1 specifically suppresses recombination that occurs independently of Rad51 and Rad54. From the recombination rates (Fig. 6d), we calculated the proportions of Rad51-dependent recombination and (Rad51-independent but) Rad52-dependent recombination at cen1 (Fig. 6e) (see the "Methods" section). In the wild-type background, almost all gene conversion occurs through Rad51-dependent recombination. However, in the pol1-R961K background, 29% of gene conversion can occur through Rad52-dependent recombination. Importantly, wild-type and pol1-R961K cells exhibited similar recombination rates (Fig. 6d), suggesting that Pol α is involved in the choice of recombination pathways at centromeres, suppressing Rad52-dependent SSA. Indeed, rad52-R45K, which impairs Rad52 SSA activity, reduced recombination in rad51Δ pol1-R961K cells (Fig. 6f). spb70-G529D and pof3-L148R also strongly increased the centromere recombination in rad51Δ and rad54Δ cells but not in rad52Δ cells

(Supplementary Fig. 4b), suggesting that Spb70 and Pof3 are also involved in the suppression of Rad52-dependent SSA at centromeres.

To determine whether Pol α suppresses Rad52-dependent SSA in non-centromeric regions, we determined gene conversion rates at the ura4 locus (Fig. 6g). In contrast to centromeres, rad51Δ and rad54Δ only partially reduced the recombination rate compared to rad52Δ, as previously observed[32]. 24% of gene conversion can occur through Rad52-dependent recombination even in wild-type cells (Fig. 6h). The sharp contrast between centromeric and non-centromeric regions in the wild-type background (Fig. 6e and h) demonstrates that Rad52-dependent recombination is suppressed specifically at centromeres. pol1-R961K increased non-centromeric recombination in wild-type, rad51Δ, and rad54Δ cells but not in rad52Δ cells (Fig. 6g), and did not change the ratio between Rad51-dependent recombination and Rad52-dependent recombination (Fig. 6h). These data indicate that, unlike centromeres, Pol α suppresses both Rad51-dependent recombination and Rad52-dependent recombination in non-centromeric regions. Replication forks might be easy to be collapsed outside centromeres.

Pol1 binds the heterochromatin protein Swi6/HP1 and facilitates transcriptional gene silencing at centromeres[53,54]. Although pol1-R961K cells were partially defective in centromeric gene silencing (Supplementary Fig. 5a), it is unlikely that Pol1 suppresses Rad52-dependent SSA through Swi6, because rad51Δ swi6Δ and rad52Δ swi6Δ cells exhibited similar levels of recombination at cen1 (Fig. 6i). It is also unlikely that

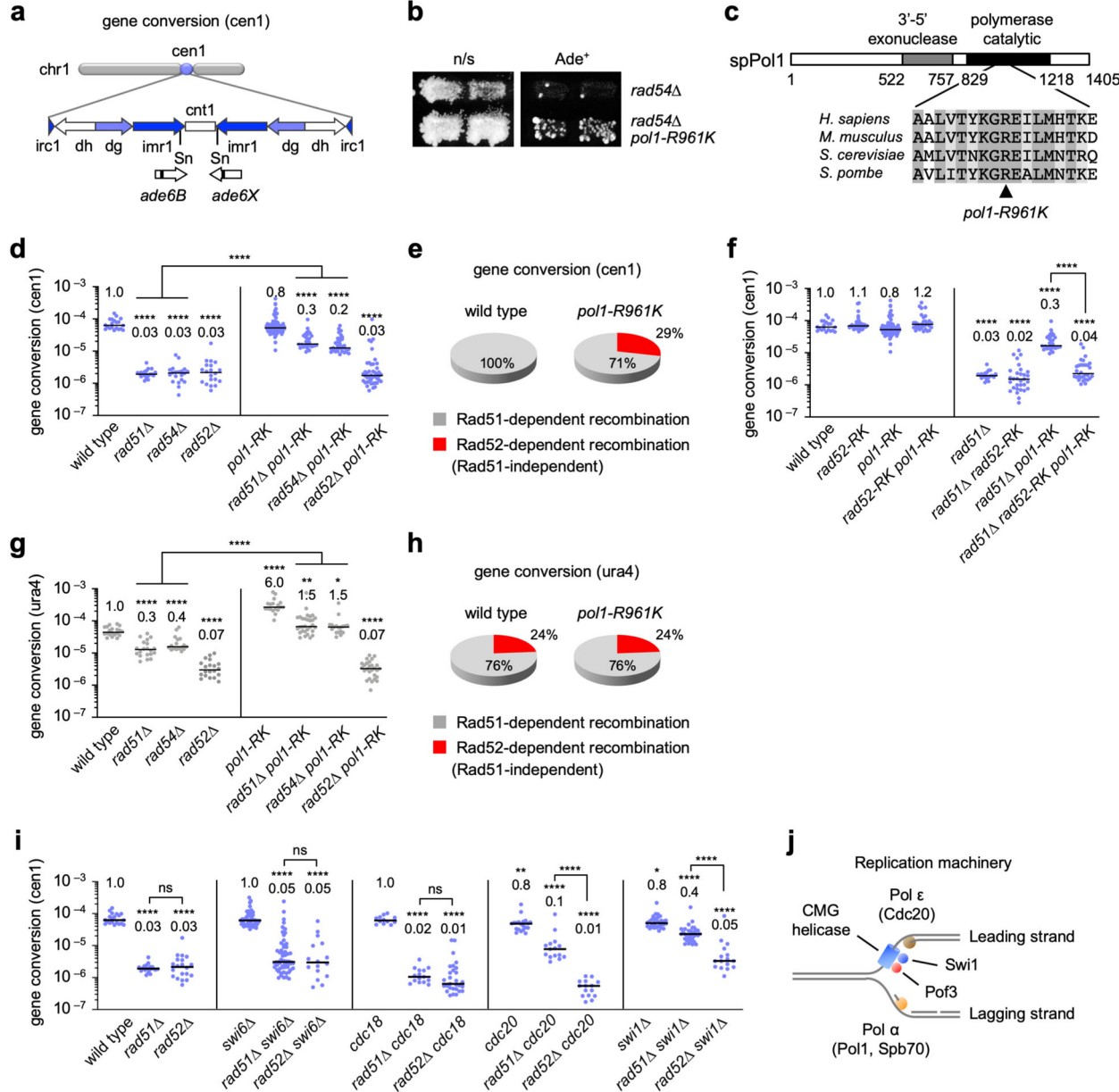

**Fig. 6 Replication elongation factors suppress Rad52-dependent SSA at centromeres. a** Illustration of *ade6B* and *ade6X* integrated at the SnaBI sites in cen1. Sn; SnaBI. **b** *rad54Δ* and *pol1-R961K rad54Δ* cells (TNF3983 and its derivative) were grown on non-selective (n/s) EMM + AL media and replicated on EMM + LG plates, on which only Ade+ recombinants grow. The cells were grown at 33 °C. **c** The *pol1-R961K* mutation in the catalytic domain of Pol1. **d** Gene conversion rates at cen1 in the wild-type, *rad51Δ*, *rad54Δ*, *rad52Δ*, *pol1-R961K*, *rad51Δ pol1-R961K*, *rad54Δ pol1-R961K*, and *rad52Δ pol1-R961K* strains (TNF3347, 3446, 3452, 3459, 4235, 4300, 4252, and 4253, respectively). **e** Pie charts showing the proportions of Rad51-dependent recombination and (Rad51-independent but) Rad52-dependent recombination at cen1 in wild-type and *pol1-R961K* cells. **f** Gene conversion rates at cen1 in the wild-type, *rad52-R45K*, *pol1-R961K*, *rad52-R45K pol1-R961K*, *rad51Δ*, *rad51Δ rad52-R45K*, *rad51Δ pol1-R961K*, and *rad51Δ rad52-R45K pol1-R961K* strains (TNF3347, 5999, 4235, 6009, 3446, 6019, 4300, and 6037, respectively). **g** Gene conversion rates at ura4 in the wild-type, *rad51Δ*, *rad54Δ*, *rad52Δ*, *pol1-R961K*, *rad51Δ pol1-R961K*, *rad54Δ pol1-R961K*, and *rad52Δ pol1-R961K* strains (TNF3631, 3635, 3645, 3643, 4215, 4371, 4378, and 4350, respectively). **h** Pie charts showing the proportions of Rad51-dependent recombination and (Rad51-independent but) Rad52-dependent recombination at ura4. **i** Gene conversion rates at cen1 in the wild-type, *rad51Δ*, *rad52Δ*, *swi6Δ*, *rad51Δ swi6Δ*, *rad52Δ swi6Δ*, *cdc18-K46*, *rad51Δ cdc18-K46*, *rad52Δ cdc18-K46*, *cdc20-M10*, *rad51Δ cdc20-M10*, *rad52Δ cdc20-M10*, *swi1Δ*, *rad51Δ swi1Δ*, and *rad52Δ swi1Δ* strains (TNF3347, 3446, 3459, 3710, 4542, 6655, 5096, 5155, 6632, 4594, 4617, 5037, 5018, 5033, and 6653, respectively). **j** Illustration of the protein components of the replication machinery. *$P < 0.05$; **$P < 0.01$; ****$P < 0.0001$; ns, non-significant $P > 0.05$. Source data for the graphs in **d**–**i** are available in Supplementary Data 1.

Pol1 suppresses Rad52-dependent SSA by promoting replication initiation. Cdc18/Cdc6 is specifically required for the initiation of DNA replication. While the *cdc18-K46* mutation causes a growth defect at 33 °C[55], *rad51Δ cdc18-K46* and *rad52Δ cdc18-K46* cells exhibited similar levels of recombination at 33 °C (Fig. 6i). *pol1-R961K* cells were no more sensitive to CPT, methyl

methanesulphonate (MMS), or hydroxyurea (HU) than wild-type cells, unlike a previously described mutant allele of *pol1*, *swi7-1*[56] (Supplementary Fig. 5b). Thus, it is unlikely that Pol1 regulates recombination through its role in repair synthesis[57].

A pair of complementary ssDNAs are prerequisite for SSA but not for Rad51-dependent recombination. The *pol1-R961K*

mutation (Fig. 6c) may impair lagging-strand synthesis and accumulate ssDNA gaps between Okazaki fragments, which are in turn used in SSA. Consistent with this idea, we observed an accumulation of spontaneous foci of Rpa2, a subunit of the RPA complex that specifically binds ssDNA, in *pol1-R961K* cells (Supplementary Fig. 6). To determine whether coordinated DNA synthesis and unwinding at replication forks suppress Rad52-dependent SSA, we examined the role of Pol ε (epsilon) and Swi1 (Fig. 6j). Pol ε is involved in leading-strand synthesis[35]; Swi1 is associated with the CMG helicase and regulates fork progression[36,37]. Like *pol1-R961K*, both *cdc20-M10* (a mutation in Pol ε catalytic subunit) and *swi1Δ* substantially increased centromere recombination in *rad51Δ* cells but not in wild-type or *rad52Δ* cells (Fig. 6i). Collectively, these results suggest that the replication machinery plays an important role in the recombination pathway choice at centromeres, probably by restricting ssDNA gap formation.

**Pol α and Swi1 suppress Rad52-dependent GCRs at centromeres.** The replication machinery may suppress centromeric GCRs, as it

prevents Rad52-dependent SSA at centromeres. Indeed, *pol1-R961K* and *swi1Δ* increased GCR rates by 12- and 15-fold, respectively (Fig. 7a), indicating that Pol1 and Swi1 suppress GCRs. It is likely that the replication machinery suppresses GCRs by promoting Rad51-dependent recombination, as neither *pol1-R961K* nor *swi1Δ* increased GCR rates in *rad51Δ* cells (Fig. 7a). *pol1-R961K* and *swi1Δ* reduced GCR rates in *rad51Δ* cells ($P = 0.0005$ and 0.0052, two-tailed Mann–Whitney test). This is probably due to increased gene conversion in *rad51Δ pol1-R961K* and *rad51Δ swi1Δ* cells (Fig. 6). Strikingly, *rad52-R45K*, that impairs SSA, reduced GCR rates in *pol1-R961K* and *swi1Δ* cells (Fig. 7b), demonstrating that Pol1 and Swi1 suppress Rad52-dependent SSA that results in GCRs. Note that *rad52-R45K pol1-R961K* and *rad52-R45K swi1Δ* cells exhibited higher GCR rates than *rad52-R45K* cells ($P < 0.0001$, two-tailed Mann–Whitney test). It is possible that, in these replication mutants, ssDNA gaps accumulate and a residual SSA activity of Rad52-R45K becomes sufficient to cause GCRs to some extent. It is also possible that, like Rad51, the replication machinery suppresses not only Rad52-dependent but also Rad52-independent GCRs (see the "Discussion" section). To determine whether the replication machinery suppresses centromeric GCRs,

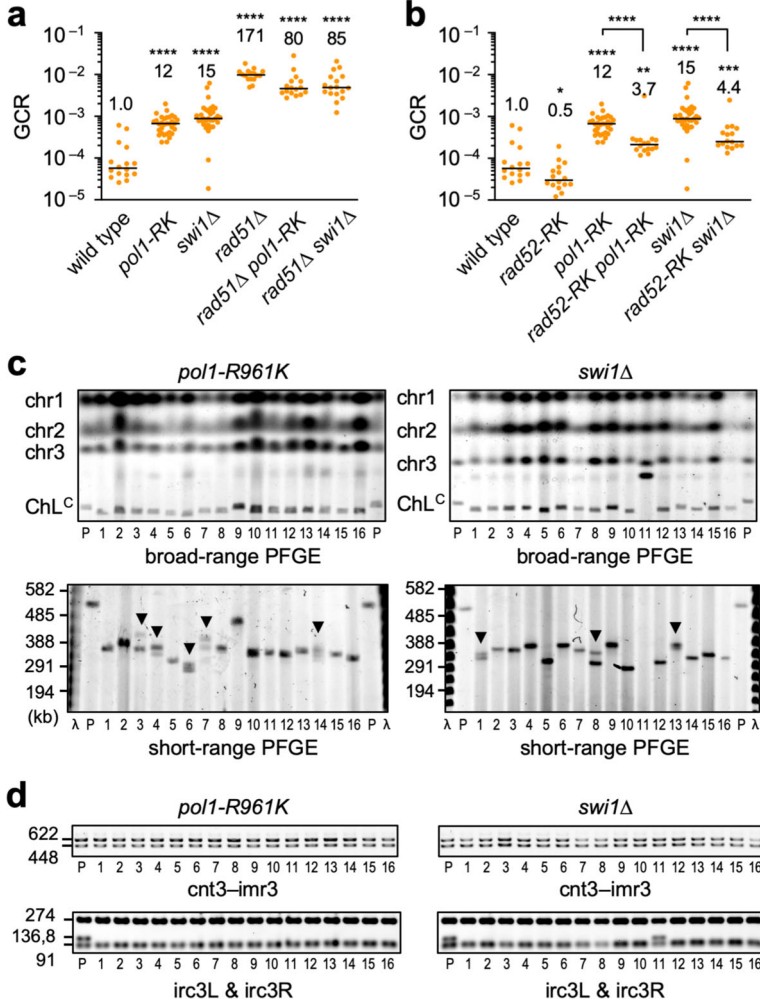

**Fig. 7 DNA Pol α and Swi1 prevent Rad52-dependent GCRs at centromeres. a** GCR rates of the wild-type, *pol1-R961K*, *swi1Δ*, *rad51Δ*, *rad51Δ pol1-RK*, and *rad51Δ swi1Δ* strains (TNF5369, 6678, 6952, 5411, 6833, and 7909, respectively). **b** GCR rates of the wild-type, *rad52-R45K* (TNF6599), *pol1-R961K*, *rad52-R45K pol1-R961K* (TNF6695), *swi1Δ*, and *rad52-R45K swi1Δ* (TNF6954) strains. *$P < 0.05$; **$P < 0.01$; ***$P < 0.001$; ****$P < 0.0001$. **c** Chromosomal DNAs from parental strains and independent GCR clones of *pol1-R961K* and *swi1Δ* were separated by broad- and short-range PFGE and stained with EtBr, as shown in Fig. 2b. Arrowheads indicate samples containing GCR products of different sizes. **d** PCR analysis of GCR products. Both sides of the cnt3–imr3 junctions (cnt3–imr3) and outermost repeats (irc3L & irc3R) were examined. Uncropped images of the gels presented in **c**, **d** are shown in Supplementary Fig. 9. Source data used for the graphs in **a**, **b** are available in Supplementary Data 1.

we performed PFGE and PCR analyses and found that most of the GCR products formed in *pol1-R961K* and *swi1Δ* cells were isochromosomes whose breakpoints were present in centromere repeats (Fig. 7c and d). These results demonstrate that the replication machinery suppresses Rad52-dependent GCRs at centromeres. Multiple ChL^C bands detected in short-range PFGE (Fig. 7c, arrowheads) suggest that Pol1 and Swi1 also suppress recombination between tandem repeats in cen3 (Fig. 2a), resulting in expansion or contraction of centromere regions.

## Discussion

Rad51-dependent recombination safeguards genome integrity. However, not much is known about how GCRs occur in Rad51-deficient cells and how they are prevented. Here, we found in fission yeast that GCRs occur through Rad52-dependent SSA at centromeres. We also showed that the Msh2–Msh3 ring-like complex and the Mus81 endonuclease act in the Rad52-dependent GCR pathway. Remarkably, replication fork components, including DNA Pol α catalytic subunit Pol1, are required for centromere-specific suppression of Rad52-dependent SSA. These data suggest that DNA replication machinery plays an important role in the recombination pathway choice at centromeres to prevent Rad52-dependent SSA that leads to GCRs.

This study provided evidence that homology-mediated GCRs occur through Rad52-dependent SSA. Loss of Rad51 increased isochromosome formation and chromosomal truncation. Loss of Rad52 eliminated ~90% of isochromosomes but did not affect truncation in *rad51Δ* cells, showing that Rad52 is specifically required for the majority of homology-mediated GCRs. To understand how Rad52 promotes GCRs, we created the *rad52-R45K* mutation that impaired the in vitro SSA activity of Rad52 protein. A recent study also showed that the R45 residue is important for the in vivo SSA between direct repeats[58]. Like *rad52* deletion, *rad52-R45K* specifically reduced isochromosomes in *rad51Δ* cells, showing that Rad52 promotes homology-mediated

GCRs through SSA. Rad52 also has an ability to promote DNA strand exchange in vitro, although not very strong[59,60]. The strand exchange can be another manifestation of the SSA activity, as the same arginine residue (R45 in fission yeast) is important for the strand exchange[41,61,62]. Rad52-dependent GCRs may occur not only at centromeres but also in other regions of chromosomes. It is unclear why GCR-prone recombination such as Rad52-dependent SSA is present from yeast to humans. It is tempting to speculate that Rad52-dependent GCRs is one of the mechanisms behind chromosomal diversity that appeared during evolution.

Our current model of how Rad52-dependent SSA promotes GCRs, with the aid of Msh2–Msh3 and Mus81, is shown in Fig. 8a. ssDNA tails are produced at spontaneous DNA damage sites. When there are no ssDNAs in the template, Rad51 catalyses DNA strand exchange with dsDNA (Fig. 8a, left). However, when ssDNA gaps are present in the template, Rad52-dependent SSA can occur between a pair of complementary ssDNAs (Fig. 8a, right). As the minimum length of DNA homology required for Rad52-dependent SSA is only 15 bp[63], Rad52 may use short ssDNA gaps produced during DNA replication and form joint molecules containing short stretches of heteroduplexes. It is possible that the Msh2–Msh3 ring-like complex[33] encircles the heteroduplex and stabilises the joint molecule. Consistent with this, it has been shown that Msh2–Msh3 is required for SSA between tandem repeats especially when the homology length is limited[47]. Physical interactions between Msh2–Msh3 and Rad52 have also been reported[64]. Mus81 forms the Mus81-Eme1 endonuclease, which preferentially cleaves joint molecules as crossovers or half-crossovers[27,48]. Rad52 enhances the DNA cleavage activity of the Mus81 complex in vitro[65,66]. Our genetic data shows that Mus81 acts in the Rad52-dependent GCR pathway. Therefore, Mus81 as well as Msh2–Msh3 may recognise joint molecules produced by Rad52-dependent SSA and cleave them as half-crossovers. Half-crossover between inverted repeats on the opposite sides of sister centromeres results in isochromosome formation[18,26].

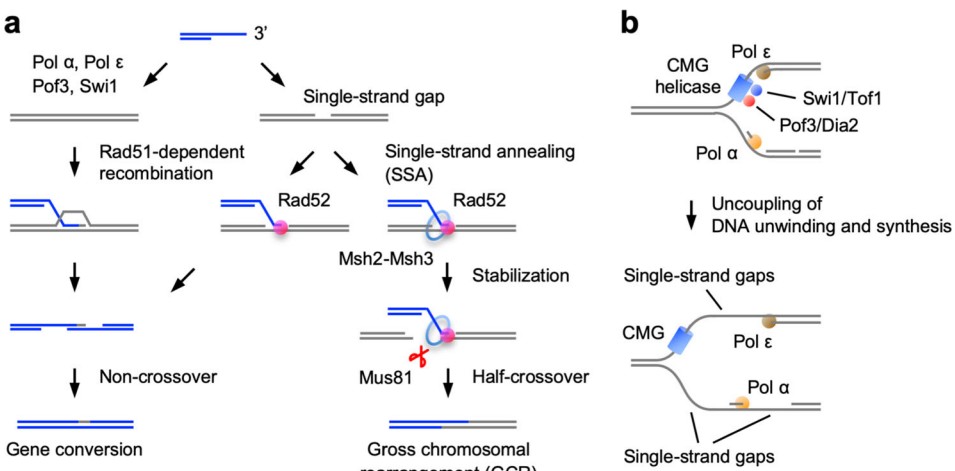

**Fig. 8 The replication machinery promotes Rad51-dependent recombination and prevents Rad52-dependent SSA at centromeres. a** The replication machinery containing Pol α, Pol ε, Pof3, and Swi1 promotes Rad51-dependent recombination by preventing single-strand gap formation at centromeres (left). However, when single-strand gaps are formed on the template DNA, Rad52-dependent SSA occurs between a pair of complementary ssDNAs (right). Specific involvement of Msh2, Msh3, and Mus81 in Rad52-dependent GCRs suggests that joint molecules formed by Rad52-dependent SSA are stabilised by the Msh2–Msh3 ring-like complex, and are resolved by the Mus81 endonuclease into half-crossover products. Half-crossover between inverted repeats that are present on the opposite sides of sister centromeres results in the formation of isochromosomes. Dissociation of the joint molecule that occurs independently of Msh2–Msh3 and Mus81 may result in gene conversion through non-crossover recombination. **b** The CMG helicase, which consists of Cdc45, MCM2-7, and GINS, is involved in the progression of replication forks. Swi1/Tof1 and Pof3/Dia2, which are associated with the CMG helicase, and lagging- and leading-strand polymerases (Pol α and Pol ε, respectively) are required for tight coupling of DNA unwinding and synthesis at centromeres. Mutations in these replication proteins can uncouple DNA unwinding and synthesis, resulting in the formation of single-strand gaps, which in turn can be used in Rad52-dependent SSA.

This study revealed that Rad52-dependent SSA facilitates two types of recombination. First, as discussed above, Rad52-dependent SSA cause half-crossover recombination that results in GCRs, with the aid of Msh2–Msh3 and Mus81 (Fig. 8a). Interestingly, loss of RAD52 has been shown to reduces cancer predisposition and increases the lifespan of adenomatous polyposis coli (APC)- or ataxia-telangiectasia mutated (ATM)-deficient animals[12,67]. Contrary to BRCA2 that promotes Rad51-dependent recombination, gene amplifications rather than mutations have been observed for *RAD52* in cancer cells[68]. These observations suggest that RAD52-dependent GCRs facilitate tumorigenesis in higher eukaryotes. Second, Rad52-dependent SSA causes conservative non-crossover recombination, independently of Msh2–Msh3 and Mus81[26] (Fig. 8a). Rad52-dependent SSA appears to be a backup of Rad51-dependent recombination. In fission yeast, *rad52* reduced gene conversion and increased DNA damage sensitivity and chromosome loss in *rad51Δ* cells. In mammals, RAD52 inactivation is synthetic lethal with *BRCA2* mutations[13]. These two opposing roles of RAD52-dependent SSA make it a potential target of chemotherapy to treat RAD51-deficient tumour, as RAD52 inactivation may specifically inhibit the tumour growth and block additional rearrangements of chromosomes.

Although Rad52-dependent SSA is the major pathway of homology-mediated GCRs in *rad51Δ* cells, there is other pathway (s) leading to homology-mediated GCRs. In *rad51Δ rad52Δ* cells, the total rate of GCRs was 33-fold higher than the wild-type level, and approximately half of the products were isochromosomes. Homology-mediated GCRs are increased by ~15-fold in *rad51Δ rad52Δ* cells compared to wild-type cells. Loss of the Rad52 paralog in fission yeast, Rti1, did not reduce GCR rates in *rad51Δ rad52-R45K* cells, suggesting that Rti1 is not involved in Rad52-independent GCR. In budding yeast, a mutation in RPA increases recombination between inverted repeats even in the absence of both Rad51 and Rad52[69], suggesting that RPA-free naked ssDNA is an important substrate for Rad52-independent GCRs. Rad52-independent SSA and/or MMEJ might be responsible for homology-mediated GCRs in *rad51Δ rad52Δ* cells, but further studies are required to understand the exact mechanism of Rad52-independent GCRs.

DNA recombination is regulated at centromeres[28–32]. Previously, we showed in fission yeast that Rad51-dependent recombination predominates at centromeres[32]. Here, we found that components of the replication machinery, Pol1 (Pol α), Spb70 (Pol α), Cdc20 (Pol ε), Pof3, and Swi1, are required to suppress Rad52-dependent SSA at centromeres. It seems that the replication machinery affects the recombination pathway choice at centromeres, to suppress Rad52-dependent SSA (in other words, to promote Rad51-dependent recombination). The *pol1-R961K* mutation in the catalytic domain of Pol α increased the proportion of Rad52-dependent SSA without changing the total rate of gene conversion at centromeres. On the other hand, in non-centromeric regions, *pol1-R961K* increased both Rad51-dependent recombination and (Rad51-independent but) Rad52-dependent recombination. Importantly, *pol1-R961K* as well as loss of Swi1, which regulates fork progression[70–72], increased Rad52-dependent isochromosome formation, demonstrating that Rad52-dependent SSA is suppressed to prevent centromeric GCRs. It will be interesting to know if similar regulation of the recombination pathway choice exists in other chromosomal regions.

How does the replication machinery affect the choice of centromeric recombination pathways to suppress Rad52-dependent SSA? A pair of complementary ssDNAs are prerequisite for SSA, but several lines of evidence suggest that limited levels of ssDNAs are formed during centromere replication. In *Xenopus* egg extracts, the replication kinetics of centromeric DNA are slow and RPA is underrepresented on centromere chromatin[73]. The Swi1 homologue in budding yeast, Tof1, slows down or pauses fork progression at centromeres[70], and loss of Swi1 increases ssDNA levels at replication forks in fission yeast and mammals[71,72]. It is possible that the replicative CMG helicase moves slowly and is tightly coupled to DNA synthesis at centromeres, resulting in limited ssDNA formation (Fig. 8b). Mutations in Swi1 or Pof3, both of which regulate fork progression[37,38,52,70–72], may uncouple DNA unwinding and synthesis, increasing single-strand gaps during replication. Mutations in Pol α or Pol ε may also increase single-strand gaps due to defects in DNA synthesis. We propose that the replication machinery restrains ssDNA gap formation at centromeres, thereby suppressing Rad52-dependent SSA at repetitive centromeres. Understanding how the replication machinery is regulated at centromeres is one of the important directions of future study.

## Methods

**Genetic procedures**. The fission yeast strains used in this study are listed in Supplementary Table 1. Standard genetic procedures were used, as previously described[26]. YE3S is YE media supplemented with 225 μg mL⁻¹ each of leucine, uracil, and adenine. To obtain mutations that increase Rad51/Rad54-independent recombination at centromeres, *rad54Δ* cells containing *ade6B/ade6X* heteroalleles at cen1 (TNF3452) were treated with 0.1 M sodium nitrate for 5–15 min and plated on non-selective YE + A plates. Independent clones were incubated as patches on YE + A plates and transferred to EMM plates. We recovered three clones that reproducibly formed elevated levels of Ade⁺ recombinants on EMM, two of which exhibited temperature-sensitive growth defects. After at least four rounds of backcrossing with the wild-type strain, we introduced a genomic library into the two clones and found that the plasmids containing *pof3* or *spb70* complemented their growth defects. Sanger sequencing of genomic DNA prepared from the two clones identified *pof3-L148R* and *spb70-G529D* mutations. For the remaining clone, genomic DNA was subjected to deep sequencing using an Illumina HiSeq 2000 platform (Eurofins Operon, Japan). Analysis of the whole-genome sequence using MAQ software (ver.0.6.6)[74] identified the *pol1-R961K* mutation. The recombination phenotypes of the three clones were complemented by plasmids containing *pof3* (pTN982), *spb70* (pTN986), and *pol1* (pTN990), respectively (Supplementary Fig. 4a).

To create the *rad52-R45K* mutant strain, PCR fragments of 1.3 and 0.7 kb were amplified using rad52-N-F1/rad52-RK-R and rad52-RK-F/rad52-C-R primers, respectively (Supplementary Table 2). Both the rad52-RK-R and rad52-RK-F primers contain the *rad52-R45K* mutation. The two overlapping PCR fragments were connected in a second round of PCR performed using rad52-N-F1 and rad52-C-R. The 1.9 kb product was introduced into the *ura4⁺:rad52⁺* strain, in which *ura4⁺* was placed upstream of *rad52⁺*. Ura⁻ transformants were selected on plates supplemented with 5-fluoroorotic acid (5FOA), and correct integration of the *rad52-R45K* mutation was confirmed by PCR and DNA sequencing.

**Purification of Rad52 protein**. Rad52-Flag and Rad52-R45K-Flag proteins were produced in the *E. coli* strain BL21(DE3), using pETDuet-1 vectors (Novagen) expressing Rad52-6His-3Flag (pTN1118) and Rad52-R45K-6His-3Flag (pTN1158), respectively. Cells were grown in 400 mL of LB media supplemented with 50 μg mL⁻¹ of ampicillin at 30 °C. At an optical density at 600 nm of ~0.5, 1 M of isopropyl-β-D-thiogalactopyranoside (IPTG) was added to a final concentration of 0.2 mM. After 4 h incubation, the cells were collected by centrifugation at 6000 × *g* for 10 min at 4 °C and stored at −80 °C. Cells were resuspended in 20 mL of buffer R (20 mM Tris-HCl pH 8.0, 500 mM NaCl, 1 mM ethylenediaminetetraacetic acid (EDTA), 10% glycerol, and 1 mM dithiothreitol (DTT)) supplemented with 1 mM phenylmethylsulphonyl fluoride (PMSF) and 2 mM benzamidine, and disrupted by nine rounds of 20 s sonication using a Sonifier 250 (Branson). After centrifugation at 35,000 × *g* for 10 min at 4 °C, an equal volume of buffer R containing 500 mM NaCl and ammonium sulphate at 60% saturation was added to the supernatant. After centrifugation at 35,000 × *g* for 30 min at 4 °C, the precipitate was recovered and suspended in 20 mL of buffer R containing 200 mM NaCl. After centrifugation at 35,000 × *g* for 30 min at 4 °C, 150 μL of Anti-FLAG M2 affinity gel (Sigma-Aldrich, A2220) was added to the supernatant. After 2 h incubation at 4 °C, the beads were washed three times with 1 mL of buffer R containing 200 mM NaCl, 12 mM MgCl₂, and 3 mM ATP, then three times with 1 mL of buffer R containing 200 mM NaCl. To elute Rad52, the beads were suspended in 300 μL of buffer R containing 200 mM NaCl and 250 ng μL⁻¹ of 3×Flag peptides and incubated at room temperature for 1 h. After recovering the supernatant, the beads were resuspended in 200 μL of the same buffer and incubated at room temperature for an additional 1 h. The eluents were combined and centrifuged at 20,400 × *g* for 10 min at 4 °C. After addition of an equal volume of buffer R without NaCl, the

supernatant was applied to a mono-Q column (GE Healthcare). The proteins were eluted at ~250 mM using a linear NaCl gradient (100 mM–1 M) in buffer R. Proteins were dialysed into storage buffer (20 mM Tris-HCl pH 8.0, 1 mM EDTA, 175 mM NaCl, 10% glycerol, and 1 mM DTT). The proteins were resolved by 12% sodium dodecyl sulphate-polyacrylamide gel electrophoresis (SDS-PAGE; acrylamide: bis-acrylamide = 37.5:1) and stained with Coomassie Brilliant Blue (CBB). The signals were analysed using ImageJ software (National Institute of Health, USA). For immunoblotting, the proteins were transferred onto Polyscreen PVDF hybridization transfer membranes (Perkin Elmer). Mouse monoclonal anti-Flag antibody (1:2000; Sigma-Aldrich, F1804) and peroxidase AffiniPure goat anti-mouse IgG (1:10,000; Jackson ImmunoResearch Laboratories) were used as primary and secondary antibodies, respectively. Blots were developed using SuperSignal West Pico chemiluminescent substrate (Thermo Fisher Scientific) and detected with ImageQuant LAS500 (GE Healthcare).

**Gel mobility shift assays**. Gel mobility shift assays were performed essentially as previously described[40]. The 5'-end of Oligo211 (5'-GAAGCATTTATCAGGGTT ATTGTCTCATGAGCGGATACATATTTGAAT-3', 48 nt) was $^{32}$P labelled using γ-$^{32}$P-ATP (Perkin Elmer, NEG002A, 3000 Ci mmol$^{-1}$) and T4 polynucleotide kinase (New England Biolabs, M0201S). The $^{32}$P-labelled Oligo211 (2 nM in DNA molecules) and Rad52 (0, 25, 50, 100, 150, or 200 nM) were mixed in 10 μL of binding buffer (32 mM Tris-HCl pH 7.8, 40 mM KCl, 0.8 mM DTT, and 0.08 mg mL$^{-1}$ bovine serum albumin (BSA; Sigma, A7906, fraction V)) and incubated at 30 °C for 10 min. After addition of 2 μL of 5× loading buffer (20 mM Tris-HCl pH 7.4, 0.5 mM EDTA, 50% glycerol, and 0.1% bromophenol blue (BPB)), the reaction mixture was applied to 10% non-denaturing PAGE (acrylamide: bis-acrylamide = 37.5:1) in 1× TAE buffer (40 mM Tris-acetate and 1 mM EDTA) at 100 V in an ice-cold tank. Gels were dried on grade DE81 ion exchange cellulose chromatography paper (Whatman, 3658-915). Radioactive signals were detected using a BAS2500 phosphorimager (Fujifilm) and quantified with Image Gauge Software version 3.4 (Fujifilm).

**Single-strand annealing (SSA) assays**. SSA assays were performed essentially as previously described[45]. Oligo508 (5'-ATTCAAATATGTATCCGCTAATGAGAC AATAACCCTGATAAATGCTTCACTAG-3', 53 nt, 0.4 nM in DNA molecules) and Rad52 (1.35 nM) were incubated in 200 μL of DNA annealing buffer (25 mM Tris-acetate pH 7.5, 100 μg mL$^{-1}$ BSA, and 1 mM DTT) at 30 °C for 10 min. Reactions were initiated by the addition of 2.4 μL of 25 nM of $^{32}$P-labelled Oligo211 to a final concentration of 0.3 nM. Aliquots (20 μL) were withdrawn at the indicated time points and mixed with 20 μL of 2× stop buffer (3% SDS, 14% glycerol, 0.2% BPB, 0.2 mg mL$^{-1}$ proteinase K (Nacalai tesque, 29442-85), and 30 nM unlabelled Oligo211), and incubated at 30 °C for 15 min. DNAs were separated by 10% non-denaturing PAGE (acrylamide: bis-acrylamide = 17:1) in 1× TBE buffer (90 mM Tris-borate pH 8.0, and 2 mM EDTA). Gels were dried and radioactive signals were detected and quantified as described for gel mobility shift assays. RPA was prepared as previously described[75,76] and used in SSA assays shown in Supplementary Fig. 3.

**GCR and chromosome loss rates**. GCR and chromosome loss rates were determined as previously described[26]. To determine the rate of Leu$^+$ Ura$^-$ Ade$^-$ GCRs, 10 mL of EMM + UA was inoculated with a single colony from an EMM + UA plate. After incubation for 2–3 d, cells were plated on YNB + UA and 5FOA + UA. After 5–9 d, the colonies formed on YNB + UA and 5FOA + UA plates were counted to determine the number of Leu$^+$ and Leu$^+$ Ura$^-$ cells, respectively. Leu$^+$ Ura$^-$ colonies formed on 5FOA + UA were streaked onto EMM + UA and then transferred to EMM + U plates to inspect adenine auxotrophy. The number of Leu$^+$ Ura$^-$ Ade$^-$ cells was obtained by subtracting the Leu$^+$ Ura$^-$ Ade$^+$ value from the Leu$^+$ Ura$^-$ value. We noticed that cells containing isochromosomes grew better than those containing translocation or truncation products.

We also attempted to determine the rate of Leu$^-$ Ura$^+$ Ade$^+$ GCRs. While 5FOA was used to select uracil auxotrophy when we determined Leu$^+$ Ura$^-$ Ade$^-$ GCR rates, no drugs like 5FOA were available to select leucine auxotrophy of Leu$^-$ Ura$^+$ Ade$^+$ GCR clones. To skirt this problem, we used the *rad51Δ* strain (TNF5411), assuming that Leu$^-$ Ura$^+$ Ade$^+$ GCRs occur at high rates as Leu$^-$ Ura$^-$ Ade$^-$ GCRs so that we can obtain Leu$^-$ Ura$^+$ Ade$^+$ clones without the need of such drugs. To determine Leu$^-$ Ura$^+$ Ade$^+$ GCR rates, 10 mL of EMM supplemented with leucine (EMM + L) was inoculated with a single colony from an EMM + L plate. After 2–3 d incubation, cells were plated on EMM + L. 200 colonies formed on EMM + L were transferred to EMM plates to inspect leucine auxotrophy. We examined 200 colonies each from 12 independent cultures (total 2,400 colonies) and found no Leu$^-$ Ura$^+$ Ade$^+$ cells. Based on these results, we estimated that Leu$^-$ Ura$^+$ Ade$^+$ GCRs occurred > 30 times less frequently than Leu$^-$ Ura$^-$ Ade$^-$ GCRs. There are ~12 and 2 copies of the dg-dh centromere repeats on the left and right sides of cen3, respectively. This asymmetry might cause a preference for Leu$^-$ Ura$^-$ Ade$^-$ GCRs over Leu$^-$ Ura$^+$ Ade$^+$ GCRs. It is also possible that Leu$^-$ Ura$^+$ Ade$^+$ GCR products are toxic to cells. Although we do not know the exact reason, our current system has a preference to detect Leu$^+$ Ura$^-$ Ade$^-$ GCRs.

To determine the rate of ChL$^C$ chromosome loss, a single colony formed on a YE3S plate was suspended in water, and the cells were plated on YE plates. After 4–6 d incubation, white and red colonies were counted. The red colonies, indicative of *ade6$^+$* loss, were transferred to EMM + UA to inspect leucine auxotrophy. The number of Leu$^-$ Ade$^-$ colonies, indicative of ChL$^C$ chromosome loss, was obtained by subtracting the Leu$^-$ Ade$^-$ value from the Ade$^-$ value. Cells were grown at 30 °C to determine GCR and chromosome loss rates. The rates of GCR and chromosome loss per cell division were calculated as described[77].

**PFGE of chromosomal DNAs**. Preparation of chromosomal DNAs and their separation by PFGE were performed as previously described[26]. Chromosomal DNAs were prepared in 0.8% low melting agarose gels (Nacalai tesque, 01161-12) and resolved using a CHEF-DRII pulsed-field electrophoresis system (Bio-Rad) under the following conditions. For broad-range PFGE, a 1600 s pulse time at 2 V cm$^{-1}$ for 42 h followed by a 180 s pulse time at 2.4 V cm$^{-1}$ for 4 h, at 4 °C in 1× TAE buffer using 0.55% Certified Megabase agarose gel (Bio-Rad, 161-3109)). For short-range PFGE, a 40–70 s pulse time at 4.2 V cm$^{-1}$ for 24 h, at 4 °C in 0.5× TBE buffer using 0.55% Certified Megabase agarose gel. DNAs were stained with 0.2 μg mL$^{-1}$ EtBr (Nacalai Tesque, 14631-94) and detected using a Typhoon FLA9000 gel imaging scanner (GE Healthcare). Gel images were processed using ImageJ software or Adobe Photoshop Elements (Adobe, San Jose, CA).

**PCR analysis of GCR products**. PCR analysis of GCR products was performed as previously described[26]. GCR products were recovered from PFGE gels using a FastGene Gel/PCR Extraction kit (Nippon Genetics, FG-91302). The cnt3–imr3 junctions were amplified using KOD FX Neo polymerase (Toyobo, KFX-201) and resolved by 1.2% Seakem GTG agarose gel (Lonza, 50070) electrophoresis in 1× TBE buffer. The irc3L and irc3R PCR products were digested with ApoI-HF (New England Biolabs, R3566L) and resolved by 1.7% Seakem GTG agarose gel electrophoresis. DNA was stained with 0.2 μg mL$^{-1}$ EtBr and detected using a Typhoon FLA9000. Sequences of PCR primers used in this assay are listed in Supplementary Table 2.

**Gene conversion between *ade6B* and *ade6X* heteroalleles**. To determine the rate of gene conversion between *ade6B* and *ade6X* at the *ura4* locus[32], 10 mL of EMM + UA was inoculated with a single colony from a YE + UA plate. After 1–3 d incubation, cells were plated on EMM + UA and EMM + UG. After 3–7 d incubation, colonies formed on EMM + UA and EMM + UG were counted to determine the number of colony-forming units and Ade$^+$ prototrophs, respectively. To determine the rate of gene conversion at cen1[26], 10 mL of EMM + A was inoculated with a single colony from a YE + A plate. After 1–3 d incubation, cells were plated onto EMM + A and EMM + G. After 3–7 d incubation, colonies formed on EMM + A and EMM + G were counted to determine the number of colony-forming units and Ade$^+$ prototrophs, respectively. Cells were grown at 33 °C. The rate of gene conversion per cell division was calculated as described[77].

The proportions of Rad51-dependent recombination and (Rad51-independent but) Rad52-dependent recombination were calculated as follows. First, we obtained the Rad51-dependent recombination rate by subtracting the median rate of recombination in the *rad51Δ* strain from that in the wild-type strain. The Rad52-dependent recombination rate was obtained by subtracting the *rad52Δ* rate from the *rad51Δ* rate. We calculated the proportion of each type of recombination from the sum of both types of recombination.

**Statistics and reproducibility**. Two-tailed Mann–Whitney tests were performed using GraphPad Prism version 8 for Mac (GraphPad Software, La Jolla, CA, USA). Two-tailed Student's *t*-tests were performed using Microsoft Excel for Mac. Two-tailed Fisher's exact tests were performed using GraphPad QuicCalcs at https://www.graphpad.com/quickcalcs/contingency1.cfm.

At least 18, 16, and 12 biologically independent experiments were performed for each strain by taking independent colonies, when we determined the rates of GCR, chromosome loss, and gene conversion, respectively. When we started yeast cultures, we picked up colonies of different sizes randomly. In the GCR assay, we recovered both large and small colonies for PFGE and PCR analyses, according to the ratio of their appearance.

**Reporting summary**. Further information on research design is available in the Nature Research Reporting Summary linked to this article.

## Data availability
The data supporting the findings of this study are available in the paper and its Supplementary Information.

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

## Acknowledgements

We thank Hisao Masukata (Osaka University, Japan) and Tatsuro S. Takahashi (Kyushu University, Japan) for critical comments on this study, and Naoko Toyofuku, Taishin Zaima, and Hirofumi Ohmori (Osaka University, Japan) for technical assistance. We also thank Shiv I. S. Grewal (National Institutes of Health, USA), Gennaro D'Urso (University of Miami, USA), and the National Bio-Resource Project of Japan for providing yeast strains and plasmids. This study was supported by Japan Society for the Promotion of Science Grant-in Aid for Scientific Research (JP23570212, JP26114711, and 18K06060, to T.N.; 18H03985 to I.H.; 19K12328 to W.K.).

## Author contributions

A.T.O., J.S., Y.K., and T.N. conceived the study and designed the experiments. A.T.O., J.S., Y.K., C.T., F.Z., and T.N. performed the experiments. A.T.O., J.S., C.T. and T.N. wrote the paper. K.A. and H.N. analysed the sequencing data. H.I. prepared the RPA. W.K. critically discussed the findings. All the authors reviewed and approved the paper.

## Competing interests

The authors declare no competing interests.
