## [Peer Review File · Communications Biology]

Reviewers' comments:

Reviewer #1 (Remarks to the Author):

A new manuscript from Nakagawa group presents an interesting discovery suggesting that annealing activity of Rad52 is responsible for the formation of isochromosomes in the absence of Rad51. Isochromosomes are chromosomes with arms that are mirror images of each other. These rearrangements are common in all organisms and the data presented here may help us understanding the exact mechanism how isochromosomes are formed.

The authors provide strong evidence that DNA binding domain of Rad52 is important for the rearrangements. Rightfully they pick arginine 45 as a residue important for the DNA binding and at the same time ssDNA annealing. This mutant similar to rad52 complete deletion suppresses formation of isochromosomes observed in rad51 cells.

The authors present that some of the mismatch repair proteins known to play a role in removal of the nonhomologous tails during single strand annealing (SSA) also play a role in formation of isochromosomes in rad51. Mus81 is shown to be important as well. Moreover they set up the screen for mutants that increase the level of recombination at centromeres and found mutants in lagging strand polymerase and Swi1 and provided evidence for their role in suppressing Rad52 mediated centromeres instability.

The manuscript is well written and overall provides interesting results.

Minor concerns:

Rad52 is not responsible for all isochromosome formation in rad51. It would be good to test rad51 rad52RK rti1 triple mutant to complete epistasis analysis in Fig. 3B,D,E. The authors suggest in discussion that remaining isochromosomes in rad52 rad51 cells can be formed by MMEJ, but perhaps weak annealing activity from Rti1 is involved. While Rti1 was shown to be expressed in meiosis only, it is still good to test it here to exclude secondary annealing activity from paralog of Rad52, Rti1.

SSA is term describing recombination between two direct repeats that leads to deletion of sequences between two repeats and one repeat. Here, SSA is a recombination that occurs between two indirect repeats converted to ssDNA in rad51 mutants that are on the opposite sites of centromere and do not lead to deletion of sequences in between. The authors should consider this problem and possibly rename their pathway or explain how their pathway differs from regular SSA.

In wild type cells elimination of arginine 45 in Rad52 results in reduction of GCR. Is this reduction (Fig 3B) corresponding to elimination of rare isochromosomes?

Are isochromosomes formed that have mirror image of the right chromosome arm carrying ura4 and ade6 genes? Are these equally frequent as leu2-CEN-leu2 isochromosomes? If not, why not?

Reviewer #2 (Remarks to the Author):

This manuscript by Onaka and investigates the contributions of several HR repair pathways to inducing instability of centromeric sequences, using fission yeast as model system. They find that in cells lacking canonical Rad51-dependent HR repair, Rad52-dependent single strand annealing is a major

factor triggering rearrangement of the centromeric region. They further provide evidence that this pathway is driven by replication fork factors, such as DNA polymerase alpha and Swi1.

While I believe that most of the experiments are well-executed and the results are interesting, I found this manuscript very hard to digest, many because the experiments are not properly introduced and discussed. As such, the general thread of the paper is hard to follow. This is a pity since I feel that the results will be of interest to the field.

Major points:

1) For many experiments, I felt asking myself: how specific is this pathway to centromeres? As far as I can tell, the authors have a non-centromeric assay (they use this in Figure 3). Can the authors also do the same analysis for the earlier experiments between rad51 and rad52?

2) In relation to this: why are these pathways specific to centromeres, as claimed by the authors? Is this really a centromere-specific pathway, or are centromeres just exquisitely sensitive to this pathway because these regions are one of the few repetitive sequences? As such, can one really consider these centromere-specific pathways? In this light, it is also important that the authors make clear, when talking about centromeres, what species they are referring to. Budding yeast and fission yeast have radically different centromeres, and please guide the reader here.

3) In general, as mention earlier, please make sure that the reader is guided through the manuscript/experiments.

minor points:

page 2, line 38-39: this is quite a strong statement, and a bit unclear. Rephrase.

page 2, line 56-58: non-crossover recombination, and gene-conversion are the same thing. In general, the authors talk about crossovers often. What do they mean with this? In many cases I feel they use this to describe a NAHR event, and this is very different from a crossover, in the classical sense. Rephrase and define.

page 2, line 61: translocations dont always lead to cell death or cancer. Rephrase.

page 3: line 67: what yeast are we talking about here?

page 3, line 77: "promoting tumorigenesis": weird way of stating this about a physiological repair pathway. rephrase.

page 3, line 80-81: not all centromeres are repetitive.

page 4, line 93: what do you mean w crossovers here?

page 5: line 129-137: the logic of this experiment is very hard to follow.

page 5, line 145: I only see 1 GCR product that is larger.

Figure 2c, 6c: can the authors include a PCR product that changes in size, as a positive control?

Reviewer #3 (Remarks to the Author):

The manuscript by Onaka et al provides a genetic analysis of gross chromosomal rearrangements (GCRs) mediated by centromere repeats in the absence of the Rad51 recombinase. The two main classes of rearrangements are isochromosomes, which result from recombination between inverted repeats flanking the centromere, and chromosome truncations. The authors show that the increased frequency of isochromosomes observed in the rad51 mutant requires the Rad52 single-strand annealing (SSA) protein. Using a rad52-R45K mutant, which is specifically impaired for annealing of single-stranded DNA oligonucleotides in vitro but not the Rad51 mediator function, they propose that SSA is the main mechanism by which GCRs are mediated in rad51 mutants. The authors then show

that Msh2, Msh3 and Mus81 are involved in Rad52-mediated GCRs. Finally, they perform a genetic screen for mutants with increased frequencies of Rad54-independent GCRs and identify components of the replication machinery that are required for Rad52-dependent SSA recombination. The authors propose a model where uncoupling of DNA unwinding and synthesis at centromeres results in the presence of single strand DNA at the replication fork leading to SSA-mediated GCRs.

Overall, this is an interesting study that provides new insights into the mechanism of Rad51-independent recombination.

Comments:

1. I found it particularly interesting that components of the replication machinery were identified as suppressors of the rad54 gene conversion defect using the ade2IR reporter integrated at the centromere because HR is usually essential when DNA synthesis is compromised (e.g., synthetic lethality between fen1 and HR mutations). Is there evidence for more recombinogenic lesions in the pol1-R961K mutant (eg., more Rpa1-GFP or Rad52-GFP foci)?
2. What is the relationship between Rad51 and the replication machinery, which are both proposed to prevent Rad52 mediated GCR? In figure 6, what is the GCR rate of the rad51 pol1-RK double mutant?
3. The frequency of GCRs is 33-fold higher in the rad51 rad52 double mutant than wild-type and half of the events are isochromosomes. Do the authors think these are Rad52 independent SSA events? There is some evidence in budding yeast for Rad52-independent SSA when RPA binding to ssDNA is compromised so the suggestion of less RPA binding in centromeric regions is attractive.
4. Because the study is done in a rad51 mutant context it is not clear if the SSA-mediated GCR events described by the authors are also relevant to WT cells. For example, msh2 or msh3 mutations seem to have no effect on GCRs in a WT background; however, rad52RK decreases GCRs by 50% in WT cells. This difference should be discussed.

Minor comments:

Figure 3b: In the rad52RK mutant, are the remaining events truncations?

Figure 4a: Why is the GCR rate in the triple mutant rad51 rad52RK msh2 higher than the double mutant rad51 rad52RK (65 vs 24)?

Figure 4c: Do the authors have an explanation for why gene conversion is not decreased by msh2 or mus81 in the rad51 background, in contrast to rad52RK?

Figures 3, 4, 5: p values should be indicated.

Figure 6a: rad52RK single mutant data should be included here along with statistical comparisons with the double mutants.

Figure 6b: what do the arrows indicate?

Line 269: effects should be affects

Line 358: dependent

Response to Referees

We gratefully thank all the reviewers for finding our study interesting. We appreciate their insightful comments, suggestions, and advice that gave us a great opportunity to improve the manuscript.

Reviewer #1 (Remarks to the Author):

A new manuscript from Nakagawa group presents an interesting discovery suggesting that annealing activity of Rad52 is responsible for the formation of isochromosomes in the absence of Rad51. Isochromosomes are chromosomes with arms that are mirror images of each other. These rearrangements are common in all organisms and the data presented here may help us understanding the exact mechanism how isochromosomes are formed.

The authors provide strong evidence that DNA binding domain of Rad52 is important for the rearrangements. Rightfully they pick arginine 45 as a residue important for the DNA binding and at the same time ssDNA annealing. This mutant similar to *rad52* complete deletion suppresses formation of isochromosomes observed in *rad51* cells.

The authors present that some of the mismatch repair proteins known to play a role in removal of the nonhomologous tails during single strand annealing (SSA) also play a role in formation of isochromosomes in *rad51*. Mus81 is shown to be important as well. Moreover they set up the screen for mutants that increase the level of recombination at centromeres and found mutants in lagging strand polymerase and Swi1 and provided evidence for their role in suppressing Rad52 mediated centromeres instability.

The manuscript is well written and overall provides interesting results.

Minor concerns:

Rad52 is not responsible for all isochromosome formation in *rad51*. It would be good to test *rad51 rad52RK rti1* triple mutant to complete epistasis analysis in Fig. 3B,D,E. The authors suggest in discussion that remaining isochromosomes in *rad52 rad51* cells can be formed by MMEJ, but perhaps weak annealing activity from Rti1 is involved. While Rti1 was shown to be expressed in meiosis only, it is still good to test it here to exclude secondary annealing activity from paralog of Rad52, Rti1.

Response:

Thank you for the insightful comments. We agree that it is good to test whether the Rad52 paralog, Rti1, plays a role in GCR in the *rad51Δ rad52-R45K* background. We created *rad51Δ rad52-R45K rti1Δ* as well as *rad52-R45K rti1Δ* strains and determined their GCR rates (**Fig. 3b**), and found that Rti1 is not essential for GCR even in the *rad51Δ rad52-R45K* and in the *rad52-R45K* backgrounds. *rti1Δ* also showed no effects on gene conversion rates (**Fig. 3d**) and CPT sensitivity (**Fig. 3e**) in both *rad51Δ rad52-R45K* and *rad52-R45K* backgrounds. We explained these results in **lines 214-219**.

“In budding yeast, the Rad52 paralog, Rad59, ... gene conversion and DNA damage repair.”

SSA is term describing recombination between two direct repeats that leads to deletion of sequences between two repeats and one repeat. Here, SSA is a recombination that occurs between two indirect repeats converted to ssDNA in *rad51* mutants that are on the opposite sites of centromere and do not lead to deletion of sequences in between. The authors should consider this problem and possibly rename their pathway or explain how their pathway differs from regular SSA.

Response:

1. As the reviewer suggested, we first explained that SSA occurs between direct repeats, and then explained that SSA can also occur between inverted repeats when a pair of complementary ssDNAs are available in **lines 77-80**.
“SSA is sometimes referred to as ... when a pair of complementary ssDNAs are available.”
2. We also explained the reason why Rad52-dependent SSA might be involved in isochromosome formation in the introduction part of Fig.1 (**lines 128-132**).
“Rad51 and Rad54 promote non-crossover recombination ... between the inverted repeats to produce isochromosomes.”

In wild type cells elimination of arginine 45 in *Rad52* results in reduction of GCR. Is this reduction (Fig 3B) corresponding to elimination of rare isochromosomes?

Response:

Yes, the *rad52-R45K* mutation reduced the GCR rate in wild-type cells, in which most of the GCR products were isochromosomes. To make this clear, we added the following sentence in **lines 199-200**.

“Rad52 may promote isochromosome formation ... GCR rates in wild-type cells (Fig. 3b).”

Are isochromosomes formed that have mirror image of the right chromosome arm carrying *ura4* and *ade6* genes? Are these equally frequent as *leu2-CEN-leu2* isochromosomes? If not, why not?

Response:

To address this concern, we performed additional experiments to detect *Leu⁻ Ura⁺ Ade⁺* clones that may contain isochromosomes that have mirror images of the right arm. However, we found that *Leu⁻ Ura⁺ Ade⁺* GCRs (*Leu⁻* GCRs) occurs > 30 times less frequently than *Leu⁺ Ura⁻ Ade⁻* GCRs (*Ura⁻ Ade⁻* GCRs). That is to say, *ade6-ura4-cen3-ura4-ade6* isochromosomes are less frequently formed than *leu2-cen3-leu2* isochromosomes. We explained these experiments and results, and briefly discussed the reason why *Leu⁻ Ura⁺ Ade⁺* GCRs are not equally frequent as *Leu⁺ Ura⁻ Ade⁻* GCRs (i.e., *leu2-cen3-leu2* isochromosome formation) in **lines 601-616**.

“We also attempted to determine the rate ... has a preference to detect *Leu⁺ Ura⁻ Ade⁻* GCR.”

Reviewer #2 (Remarks to the Author):

This manuscript by Onaka and investigates the contributions of several HR repair pathways to inducing instability of centromeric sequences, using fission yeast as model system. They find that in cells lacking canonical Rad51-dependent HR repair, Rad52-dependent single strand annealing is a major factor triggering rearrangement of the centromeric region. They further provide evidence that this pathway is driven by replication fork factors, such as DNA polymerase alpha and Swi1. While I believe that most of the experiments are well-executed and the results are interesting, I found this manuscript very hard to digest, many because the experiments are not properly introduced and discussed. As such, the general thread of the paper is hard to follow. This is a pity since I feel that the results will be of interest to the field.

Response:

We are very sorry for the confusion that we caused. Our data suggest that Rad52-dependent SSA and GCR are suppressed (not driven) by replication fork factors. As the reviewer pointed out, this kind of confusion was caused because the experiments were not properly introduced and discussed. So, we reconsidered the whole text thoroughly and improved many parts, as described below. We appreciate all the comments from this reviewer, because they helped us to improve this manuscript.

Major points:

1) For many experiments, I felt asking myself: how specific is this pathway to centromeres? As far as I can tell, the authors have a non-centromeric assay (they use this in Figure 3). Can the authors also do the same analysis for the earlier experiments between *rad51* and *rad52*?

Response:

As the reviewer suggested, we performed the non-centromeric assay (**Fig. 6g and h**) and also revised the following parts.

1. At centromeres, Rad51-dependent recombination mainly occurs and (Rad51-independent but) Rad52-dependent recombination is underrepresented in the wild-type background (Fig. 6d). To make it clear, we created the pie charts (**Fig. 6e**) that show the proportions of Rad51-dependent recombination and (Rad51-independent but) Rad52-dependent recombination at *cen1*. In the wild-type background, almost all gene conversion occurs through Rad51-dependent recombination (Fig. 6e). However, in the *pol1-R961K* background, 29% of gene conversion can occur through Rad52-dependent recombination (Fig. 6e). Importantly, *pol1-R961K* did not increase the recombination rate in the wild-type background (Fig. 6d), showing that Pol1 specifically affects the ratio between Rad51-dependent and Rad52-dependent recombination. We explained these results in **lines 298-312**.
“To see whether Pol1 suppresses ... recombination in *rad51Δ pol1-R961K* cells (Fig. 6f).”
2. As the reviewer suggested, we determined the recombination rate at the *ura4* locus (a non-centromeric assay) (**Fig. 6g and h**). At this non-centromeric locus, *rad51Δ* and *rad54Δ* only partially reduced recombination as compared to *rad52Δ* (**Fig. 6g**), and the 24% of gene conversion can occur through Rad52-dependent recombination (**Fig. 6h**). Comparison of the recombination profile at *cen1* (**Fig. 6e**) and *ura4* (**Fig. 6h**) in wild-type cells indicates that

Rad52-dependent recombination is specifically suppressed at centromeres. At the *ura4* locus, *pol1-R961K* increased the recombination not only in *rad51Δ* and *rad54Δ* but also in wild-type backgrounds (**Fig. 6g**), and it did not change the ratio between Rad51-dependent recombination and Rad52-dependent recombination (**Fig. 6h**), showing that *pol1-R961K* increases both types of recombination at a non-centromeric locus. Collectively, these results show that Pol α is required for centromere-specific suppression of Rad52-dependent SSA. On the other hand, Pol α is not involved in the choice of recombination pathways outside centromeres. We explained these results in **line 317-330**.

“To see whether, as is the case in centromeres ... collapse and induce both types of recombination.”

3. Figs. 1, 2, 3, and 4 show that Rad52-dependent single-strand annealing (SSA) is required for GCRs at centromeres. However, it is possible that Rad52-dependent SSA is also responsible for homology-mediated GCRs outside centromeres. So, we added the following sentence in the Discussion (**lines 405-407**).

“Rad52-dependent GCR may occur not only ... repetitive sequences are present”

2) In relation to this: why are these pathways specific to centromeres, as claimed by the authors? Is this really a centromere-specific pathway, or are centromeres just exquisitely sensitive to this pathway because these regions are one of the few repetitive sequences? As such, can one really consider these centromere-specific pathways? In this light, it is also important that the authors make clear, when talking about centromeres, what species they are referring to. Budding yeast and fission yeast have radically different centromeres, and please guide the reader here.

Response:

1. We discussed how the replication machinery suppresses Rad52-dependent SSA at centromeres, in the last paragraph of the Discussion (**lines 478-493**).
“How does the replication machinery affect ... important directions of future study.”
2. As the review suggested, we explained the difference of centromeres in budding yeast and fission yeast in the Introduction (**lines 87-89**).
“Many organisms, including humans ... consists of non-repetitive sequences.”

3) In general, as mention earlier, please make sure that the reader is guided through the manuscript/experiments.

Response: To guide the reader through the manuscript, we made following changes.

1. We revised the **Abstract**.
2. We explained the difference between crossover and half crossover recombination in the Introduction (**lines 59-62**).
“In crossover and half crossover ... cleavage of joint molecules.”
3. We explained the reason why it is important to understand how the recombination pathway is chosen at centromeres (**lines 103-109**).

“In fission yeast, recombination occurs at ... recombination predominates at centromeres.”

4. We explained the reason why Rad52-dependent SSA might cause isochromosome formation in **lines 125-132**.
“Rad51, Rad52, and Rad54 are essential for ... the inverted repeats to produce isochromosomes.”
5. We discussed how chromosomal truncation is produced independently of Rad52 in **lines 180-183**.
“On the other hand, *rad52* Δ did not ... telomerase activity at damage sites.”
6. We revised the section title to clarify the main conclusion (**line 186**), because it was rather descriptive.
“Rad52-dependent SSA activity is required for homology-mediated GCR”
7. We revised an explanation in the Results (**lines 205-208**).
“While no significant effects ... gene conversion that occurs independently of Rad51.”
8. We revised an explanation in the Results (**lines 242-243**).
“Collectively, these results demonstrate ... homology-mediated GCR.”
9. Because Fig. 3 became overloaded after addition of new data, we divided the original Fig. 3 into two Figs. We separated the genetic data and the biochemical data into **Fig. 3** and a new **Fig. 4**, respectively.
10. To explain how Msh2, Msh3, Msh6, and Mlh1 are involved in single-strand annealing (SSA) and DNA mismatch repair (MMR), we created **an illustration** in **Fig. 5a**.
11. To explain how Mus81 may produces half-crossovers, we created **an illustration** in **Fig. 5b**.
12. To make it clear that the *pol1-R961K* mutation changes the evolutionary conserved amino acid in the catalytic domain of Pol α , we created a new panel **Fig. 6c** and explained that in **lines 289-291**.
“the *pol1-R961K* mutation ... suggesting a defect in DNA synthesis.”
13. We revised an explanation of Fig. 6d (**lines 299-301**).
“Loss of either Rad51, Rad54, or Rad52 ... SSA hardly occurs at centromeres.”
14. We explained the idea behind the following experiments in **lines 343-345**.
“A pair of complementary ... which are in turn used in SSA.”
15. We explained the idea behind the following experiments in **lines 347-350**.
“To determine whether the coordinated ... replication machinery components (Fig. 6j).”
16. We discussed how replication machinery suppresses Rad52-dependent SSA in **lines 356**.
“probably by restricting ssDNA gap formation.”

17. We explained the idea behind the following experiments in **lines 358-360**.
“As the replication machinery prevents ... it may also suppress centromeric GCRs.”
18. We revised a model shown in **Fig. 8a** and added “Non-crossover” in the Rad51-dependent recombination pathway to make clear how gene conversion and GCR are different.
19. We asked an English language service to get our manuscript copy-edited.

minor points:

page 2, line 38-39: this is quite a strong statement, and a bit unclear. Rephrase.

Response:

We rephrased the statement (**lines 37-38**).

“Homologous recombination ... gross chromosomal rearrangements (GCRs).”

page 2, line 56-58: non-crossover recombination, and gene-conversion are the same thing. In general, the authors talk about crossovers often. What do they mean with this? In many cases I feel they use this to describe a NAHR event, and this is very different from a crossover, in the classical sense. Rephrase and define.

Response:

We rephrased the explanation and defined non-crossover recombination, crossover recombination, and break-induced replication in the Introduction (**lines 55-63**).

“This is true for recombination that occurs ... template DNA to the chromosome end.”

page 2, line 61: translocations dont always lead to cell death or cancer. Rephrase.

Response:

We rephrased the sentence in **lines 63-64**.

“GCRs such as translocations can cause cell death or genetic diseases including cancer^{1, 2}.”

page 3: line 67: what yeast are we talking about here?

Response:

We made it clear that the statement is true for both fission yeast and budding yeast in **lines 69-70**.

“In both fission yeast and budding yeast, ... Rad51-dependent recombination⁴.”

page 3, line 77: "promoting tumorigenesis": weird way of stating this about a physiological repair pathway. rephrase.

Response:

We rephrased the sentence in **lines 82-84**.

“In addition to their roles in DNA damage repair, ... to Rad51-dependent recombination¹¹⁻¹⁴.”

page 3, line 80-81: not all centromeres are repetitive.

Response:

We rephrased the sentences in **lines 85-89**.

“However, centromeres can be vulnerable to ... consist of non-repetitive sequences.”

page 4, line 93: what do you mean w crossovers here?

Response:

We rephrased the sentences in **lines 97-101**.

“Previously, we have shown that Rad51 and Rad54 ... GCR event in the absence of Rad51.”

page 5: line 129-137: the logic of this experiment is very hard to follow.

Response:

We rewrote the text to make the logic of the experiments clear in **lines 147-155**.

“Rad52-dependent SSA may act as ... cells are defective in Rad51-dependent recombination.”

page 5, line 145: I only see 1 GCR product that is larger.

Response:

There is another GCR product that is larger than the parental in Supplementary Fig. 1a. To make it clear, we added the sample number (#20) of the second one (**lines 165-166**).

“(Fig. 2b, sample #12; Supplementary Fig. 1a, sample #20; and Table 1).”

Figure 2c, 6c: can the authors include a PCR product that changes in size, as a positive control?

Response:

1. When we amplified left and right sides of the cnt3–imr3 junctions, we always detected two PCR bands of 448 and 622-bp (Figs. 2c and 7d (6c in the previous version)). So, we could not include a PCR product of other size. As a substitute, we performed the cnt3–imr3 PCR using a pair of primers that only amplify one side of the cnt3–imr3 junctions, to make sure that the PCR fragments of 448 and 622 bp are amplified from the left and the right side of the cnt3–imr3 junctions, respectively. We showed the results of this control experiment in **Supplementary Fig. 1c** and referred to it in **line 172**.

“(Fig. 2c and Supplementary Figs. 1b and c)”.

2. We also indicated relevant lengths of DNA fragments in standard DNA ladders on the left of the wild-type panels in **Supplementary Fig. 10b**.

Reviewer #3 (Remarks to the Author):

The manuscript by Onaka et al provides a genetic analysis of gross chromosomal rearrangements (GCRs) mediated by centromere repeats in the absence of the Rad51 recombinase. The two main classes of rearrangements are isochromosomes, which result from recombination between inverted repeats flanking the centromere, and chromosome truncations. The authors show that the increased frequency of isochromosomes observed in the *rad51* mutant requires the Rad52 single-strand annealing (SSA) protein. Using a *rad52-R45K* mutant, which is specifically impaired for annealing of single-stranded DNA oligonucleotides in vitro but not the Rad51 mediator function, they propose that SSA is the main mechanism by which GCRs are mediated in *rad51* mutants. The authors then show that Msh2, Msh3 and Mus81 are involved in Rad52-mediated GCRs. Finally, they perform a genetic screen for mutants with increased frequencies of Rad54-independent GCRs and identify components of the replication machinery that are required for Rad52-dependent SSA recombination. The authors propose a model where uncoupling of DNA unwinding and synthesis at centromeres results in the presence of single strand DNA at the replication fork leading to SSA-mediated GCRs.

Overall, this is an interesting study that provides new insights into the mechanism of Rad51-independent recombination.

Comments:

1. I found it particularly interesting that components of the replication machinery were identified as suppressors of the *rad54* gene conversion defect using the *ade2IR* reporter integrated at the centromere because HR is usually essential when DNA synthesis is compromised (e.g., synthetic lethality between *fen1* and HR mutations). Is there evidence for more recombinogenic lesions in the *pol1-R961K* mutant (eg., more Rpa1-GFP or Rad52-GFP foci)?

Response:

Thank you for the suggestion. We detected spontaneous focus formation of Rpa2-mCherry and found that the *pol1-R961K* mutation increased cells containing Rpa2 foci, suggesting that *pol1-R961K* accumulates single-stranded DNA. The data are presented in the new **Supplementary Fig. 6**, and are explained in **lines 343-347**.

“A pair of complementary ssDNAs ... specifically binds ssDNA (Supplementary Fig. 6).”

2. What is the relationship between Rad51 and the replication machinery, which are both proposed to prevent Rad52 mediated GCR? In figure 6, what is the GCR rate of the *rad51 pol1-RK* double mutant?

Response:

Thank you for the insightful comment. We created *rad51Δ pol1-R961K* as well as *rad51Δ swi1Δ* double mutants and determined their GCR rates (**Fig. 7a** (Fig. 6 in the previous version)). We found that neither *pol1-R961K* nor *swi1Δ* increased GCRs in *rad51Δ* cells, suggesting that the replication machinery and Rad51 suppress GCRs in the same pathway. The results are explained in **lines 363-368**.

“Consistent with the idea that the replication...*rad51Δ pol1-R961K* and *rad51Δ swi1Δ* cells (Fig. 6).”

3. The frequency of GCRs is 33-fold higher in the *rad51 rad52* double mutant than wild-type and half of the events are isochromosomes. Do the authors think these are Rad52 independent SSA events? There is some evidence in budding yeast for Rad52-independent SSA when RPA binding to ssDNA is compromised so the suggestion of less RPA binding in centromeric regions is attractive.

Response:

Thank you for the insightful comments. We do not know the exact mechanism of Rad52-independent GCR. But, during the revision, we found that *Rti1* is dispensable for GCR. We also found the paper showing that, in budding yeast, a mutation in RPA increases recombination between inverted repeats even in the absence of both Rad51 and Rad52 (Mott and Symington, 2011). We created a new paragraph discussing how Rad52-independent GCR might occur in **lines 448-459**.

“Although Rad52-dependent SSA is ... how Rad52-independent GCR occurs.”

4. Because the study is done in a *rad51* mutant context it is not clear if the SSA-mediated GCR events described by the authors are also relevant to WT cells. For example, *msh2* or *msh3* mutations seem to have no effect on GCRs in a WT background; however, *rad52RK* decreases GCRs by 50% in WT cells. This difference should be discussed.

Response:

We added the *rad52-R45K* and *rad51Δ rad52-R45K* data in **Fig. 5a** and discussed the difference between *rad52-R45K* and *msh2/3Δ* in **lines 254-260**.

“*msh2Δ* and *rad52-R45K* did not additively ... a supplementary role in Rad52-dependent GCR.”

Minor comments:

Figure 3b: In the *rad52RK* mutant, are the remaining events truncations?

Response:

We performed detailed analysis of GCR products that are formed in *rad52-R45K* and *rad51Δ rad52-R45K* cells (**Supplementary Fig. 2**). In *rad52-R45K* cells, all the GCR products examined were isochromosomes but not truncations, showing that the *rad52-R45K* mutation does not induce chromosomal truncation by itself. In *rad51Δ rad52-R45K* cells, like *rad51Δ rad52Δ* cells, around half of the GCR products examined were isochromosomes and the rest of them were truncation products. We included the numbers of each GCR products detected in *rad52-R45K* and *rad51Δ rad52-R45K* cells into **Table 1**. These results show that Rad52-dependent SSA is specifically required for isochromosome formation. We explained these results in **lines 194-196**.

“However, like *rad52Δ*, *rad52-R45K* reduced ... (Table 1; Supplementary Fig. 2).”

We also revised a sentence in the Discussion (**lines 400-402**).

“Like *rad52* deletion, *rad52-R45K* specifically reduced ... through Rad52-dependent SSA”

Figure 4a: Why is the GCR rate in the triple mutant *rad51 rad52RK msh2* higher than the double mutant *rad51 rad52RK* (65 vs 24)?

Response:

We discussed the GCR rate of the *rad51Δ rad52-R45K msh2Δ* triple mutant in **lines 260-262**.
“We also noticed that *msh2Δ* increased GCR ... GCRs in the *rad51Δ rad52-R45K* background.”

Figure 4c: Do the authors have an explanation for why gene conversion is not decreased by *msh2* or *mus81* in the *rad51* background, in contrast to *rad52RK*?

Response:

We added the *rad52-R45K* and *rad51Δ rad52-R45K* data in **Fig. 5c** and explained the difference between *rad52-R45K* and *msh2Δ* and *mus81Δ* in **lines 270-275**.

“We also examined whether, like Rad52 ... the Rad52-dependent GCR pathway.”

We also explained a possible reason why *msh2Δ* or *mus81Δ* did not reduce gene conversion in the *rad51Δ* background in **lines 428-431**.

“Msh2-Msh3 as well as Mus81 may ... gene conversion through non-crossover recombination²⁶.”

Figures 3, 4, 5: p values should be indicated.

Response:

As the reviewer suggested, we indicated the *P* values in the legend of **Figs. 3, 5, 6, and 7** (Figs. 3, 4, and 5 in the previous version).

Figure 6a: *rad52RK* single mutant data should be included here along with statistical comparisons with the double mutants.

Response:

As the reviewer suggested, we included the *rad52-R45K* data in **Fig. 7b** (Fig 6a in the previous version), and statistically compared GCR rates of *rad52-R45K* cells with those of *pol1-R961K rad52-R45K* and *swi1Δ rad52-R45K* cells in the text (**lines 370-375**).

“Note that *rad52-R45K pol1-R961K* ... but also Rad52-independent GCR.”

Figure 6b: what do the arrows indicate?

Response:

We explained what the arrowheads indicate in the legend of **Fig. 7** (Fig 6a in the previous version).

“Arrowheads indicate samples containing GCR products of different sizes.”

Line 269: effects should be affects

Response:

Thank you for spotting this error. We have modified the sentence (**line 331**).

Line 358: dependent

Response:

We have corrected the error (**line 469**).

REVIEWERS' COMMENTS:

Reviewer #1 (Remarks to the Author):

The revised manuscript is improved. My concerns were fully addressed. Perhaps the authors could refer to previous manuscript describing the role of R45 of rad52 in ssDNA annealing in fission yeast PMID: 31542296.

Reviewer #2 (Remarks to the Author):

The authors have adequately addressed the concerns/questions raised by the reviewer in the revised manuscript. I recommend publication of this interesting study.

Reviewer #3 (Remarks to the Author):

The authors have addressed all of my concerns.

Response to Referees

Reviewer #1 (Remarks to the Author):

The revised manuscript is improved. My concerns were fully addressed. Perhaps the authors could refer to previous manuscript describing the role of R45 of rad52 in ssDNA annealing in fission yeast PMID: 31542296.

Response:

As suggested, we referred to a recent manuscript (PMID: 31542296, Yan et al., 2019) that describes a role of Rad52-R45 in single-strand annealing (SSA) between tandem repeats in **lines 382-383**.

“A recent study also showed that the R45 residue ... *in vivo* SSA between direct repeats⁵⁸.”

Reviewer #2 (Remarks to the Author):

The authors have adequately addressed the concerns/questions raised by the reviewer in the revised manuscript. I recommend publication of this interesting study.

Reviewer #3 (Remarks to the Author):

The authors have addressed all of my concerns.